# Predicting wavelength-dependent photochemical reactivity and selectivity

Jan P. Menzel [1,2,3], Benjamin B. Noble[4], James P. Blinco [1,2✉] & Christopher Barner-Kowollik [1,2✉]

Predicting the conversion and selectivity of a photochemical experiment is a conceptually different challenge compared to thermally induced reactivity. Photochemical transformations do not currently have the same level of generalized analytical treatment due to the nature of light interaction with a photoreactive substrate. Herein, we bridge this critical gap by introducing a framework for the quantitative prediction of the time-dependent progress of photoreactions via common LEDs. A wavelength and concentration dependent reaction quantum yield map of a model photoligation, i.e., the reaction of thioether $o$-methylbenzaldehydes via $o$-quinodimethanes with $N$-ethylmaleimide, is initially determined with a tunable laser system. Combined with experimental parameters, the data are employed to predict LED-light induced conversion through a wavelength-resolved numerical simulation. The model is validated with experiments at varied wavelengths. Importantly, a second algorithm allows the assessment of competing photoreactions and enables the facile design of $\lambda$-orthogonal ligation systems based on substituted $o$-methylbenzaldehydes.

[1] School of Chemistry and Physics, Queensland University of Technology (QUT), Brisbane, QLD, Australia. [2] Centre for Materials Science, Queensland University of Technology (QUT), Brisbane, QLD, Australia. [3] Centre for Data Science, Queensland University of Technology (QUT), Brisbane, QLD, Australia. [4] School of Engineering, College of Science, Engineering and Health, RMIT University, Melbourne, VIC, Australia. ✉email: j.blinco@qut.edu.au; christopher.barnerkowollik@qut.edu.au

Photochemistry is undergoing a renaissance through adopting tunable lasers and light-emitting diodes as tools to perform light-induced reactions. However, the transformation toward using photons as the reagents of the 21st century is in its infancy, with a number of synthetic fields only just starting to reap the benefits of precision photochemistry to its full extent[1]. This paradigm change is not only of academic interest for synthetic or biomedical photochemistry and photopharmacology, but has critical connotations for industrial applications[2–5]. The opportunity to take advantage of the properties of light sources for improved photochemical outcomes is important in all fields of photochemistry, as was also highlighted in a recent review on photo-catalysis[6].

Narrow near monochromatic or monochromatic emission spectra, combined with a deeper understanding of photoreactivity has led to the design of a plethora of advanced synthetic methods, which take advantage of the properties of the chromophores and their interaction with light of specific wavelengths[7–12]. Of particular interest within additive manufacturing are combinations of photoreactions that allow the highly specific selection of dual reaction channels dependent solely on the color of light[13,14]. Addressing such challenges requires an in-depth understanding of photokinetic behavior, i.e., the rationalization of how fast light-induced reactions proceed under defined conditions and their associated dependence on experimental parameters. Attempts to predict photochemical kinetics have been reported, yet the resulting methods have not been widely utilized[15,16]. While there

are—occasionally—reports of reaction quantum yields, even with their knowledge, the prediction of photochemical conversion is seldomly performed. Constraints in predictive models and the lack of widely applicable methods call for the development of a platform technology for the prediction of photokinetic traces. An easy-to-use, inexpensive, rapid, and accessible methodology for accurate predictions will unlock the full potential of photochemical reactions to complement their thermal analogs.

Herein, we introduce a comprehensive method based on the fusion of monochromatic reaction data and numerical simulation for the prediction of irradiation time-dependent conversion using a model photochemical ligation reaction. Our conceptual approach is outlined in Fig. 1, middle and right panel.

Specifically, we initially identify and measure all required parameters, establish a controlled, reproducible irradiation setup employing light-emitting diodes and finally arrive at a quantitative prediction of photochemical reaction kinetics at different colors of light through numerical simulation.

Photochemists select an irradiation source considering the Grotthus–Draper law, also referred to as the first law of photochemistry[17]: A photochemical reaction can only proceed, if at least a fraction of the emitted light is of a wavelength that is absorbed by the substrate. Under the precision photochemistry paradigm, the emission spectrum of the light source needs to be determined and reported to allow for further calculations and reproducibility. Any investigation into the reaction kinetics of a photoreaction or a prediction shall also be conducted with awareness of the

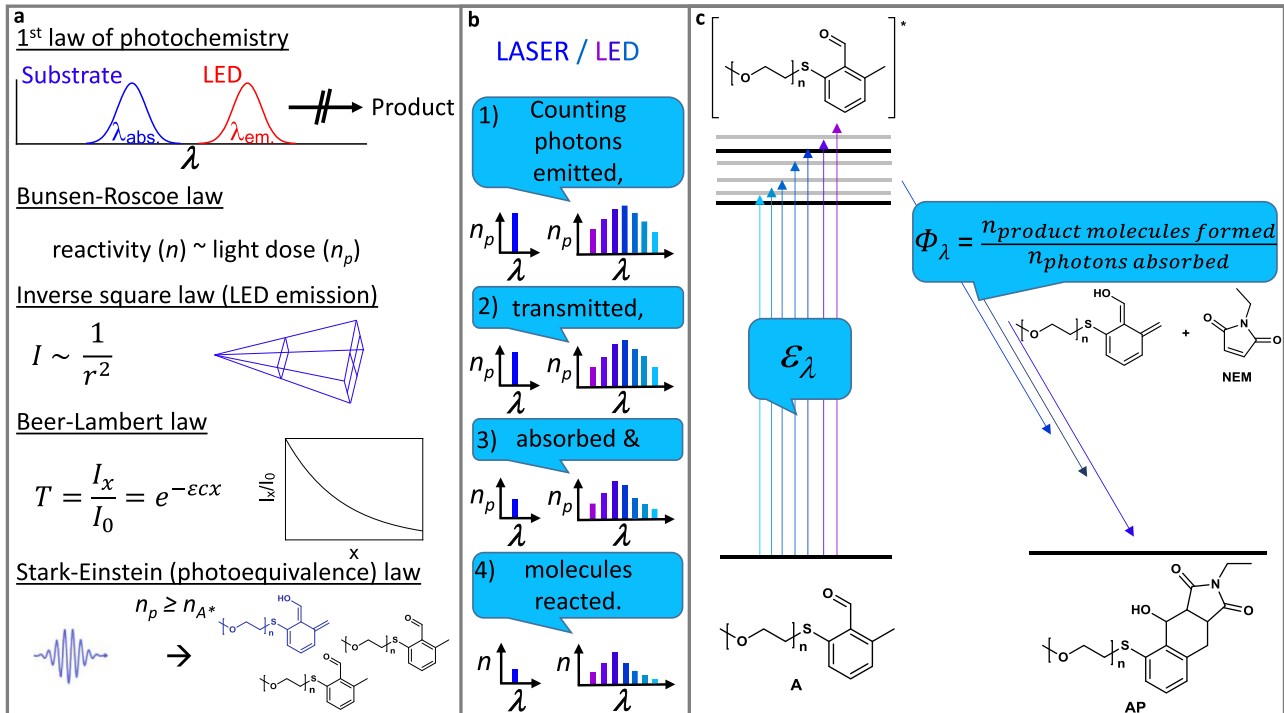

**Fig. 1 From photochemical laws to precision photochemistry. a** Visualization of laws relevant to precision photochemistry. **b** The concept of precision photochemistry with lasers and LEDs: (1) The light source needs to be characterized quantitatively in a wavelength ($\lambda$) resolved fashion. (2) Loss of photons during transmittance at the reaction vessel needs to be considered. (3) Absorption (molar attenuation coefficient $\varepsilon_\lambda$) and competing absorption need to be quantified. (4) The reaction quantum yield ($\Phi_\lambda$) determines the amount of formed product molecules. Where relevant, the number of photons $n_p$ and the number of formed product molecules $n$ needs to be quantified—in case of a polychromatic light source this needs to be quantified in a wavelength-resolved way. **c** A simplified Jablonski diagram for the photoenol ligation of poly(ethylene glycol) bound thioether o-methylbenzaldehyde A ($\alpha$-methyl-$\omega$-[(2-formyl-3-methylphenyl)thio]-poly(ethylene glycol)) with N-ethylmaleimide NEM. The molar attenuation coefficient is required to determine the number of excited states and the distribution of initially formed vibrationally and electronically excited states. Intermediate steps (intersystem crossing, bond rotations, H-shift) are omitted for clarity. The reaction quantum yield reported in this work is defined by the number of product molecules AP ($\alpha$-methyl-$\omega$-[2-ethyl-4-hydroxy-1,3-dioxo-2,3,3a,4,9,9a-hexahydro-1H-benzo[f]isoindol-5-yl)thio]-poly(ethylene glycol)) divided by the number of photons absorbed by A.

Bunsen–Roscoe law[17], i.e., the notion that observed reactivity is proportional to the dose of irradiated light[1]. Any change to the reaction setup may also lead to a varied reaction time. Practically, this may imply that further to power measurements, the (wavelength-dependent) transmittance of the reaction vessel can play a decisive role in determining the dose of light delivered to the recipient molecules. Light absorption is governed by the Beer–Lambert law, allowing the calculation of the degree of light absorption by the reactive (or reaction-causing) chromophore as well as potentially competing absorption of molecules, which absorb light of the respective wavelength without leading to product formation. Required for the application of the Beer–Lambert law is the wavelength-dependent molar attenuation coefficient ($\varepsilon_\lambda$) of the light-absorbing species, possibly both reactants and products. Finally, when all these parameters are considered, the otherwise most important measure of photochemical reactivity, the (potentially wavelength and concentration-dependent) reaction quantum yield ($\Phi_{\lambda, c}$), needs to be determined. Its importance is reflected in the Stark–Einstein law, also referred to as the photo-equivalence law, because for each photon—each quantum—only one primary photochemical event can occur. An overview of the above-mentioned laws is shown in Fig. 1, left panel.

The ability to predict product molecule numbers—conversion—rests on the ability to quantify light and its interaction with the photoreactor setup and sample in terms of photon numbers with application of all the above-mentioned laws. Yet, importantly, reproducibility of time-dependent conversion in the case of most common light sources is only possible when the inverse-square law is considered. If the emitted light is not collimated, the exact distance and geometry between the light source and irradiated sample is critical in determining the delivered light dose and thus, the kinetics of the photoreaction.

Therefore, we designed a 3D printed LED batch precision photoreactor, ensuring that photoreactions can be carried out reproducibly, as the design defines the relative position of LED and sample glass vial. Through a closely related design of a 3D printed detector scaffold (refer to Supplementary Information, section 1.4), a power measurement with this scaffold allows determining the light dose that reaches the sample vial inside the respective reactor. While alternative and higher-powered photoreactors are commercially available and in use in academic research[18–20], the approach to 3D-print a custom photoreactor scaffold has distinct advantages: First, the design is made to suit the glass vials that are used here, enabling photoreactions at particularly small solution volumes. Second, any commercially available LED and thus available wavelength can be used, whereas some commercial photoreactors are limited to a few predefined wavelengths. Third, 3D printing is extremely cost-effective in comparison to buying commercially available photoreactors, and 3D printing services are generally available. Most importantly, however, the corresponding detector scaffold allows facile determination of the light dose in the reactor, which is important as individual LEDs, even those of the same color can vary considerably in their output, which depends for example on the temperature of the semiconductor and thus cooling efficiency. Selected for experimental testing of the numerical prediction of photoreactivity is the photoenol ligation of thioether o-methylbenzaldehyde A (α-methyl-ω-[(2-formyl-3-methylphenyl)thio]-poly(ethylene glycol)) with N-ethylmaleimide NEM via o-quinodimethanes to the Diels-Alder ligation product AP[21], as shown in Fig. 1. With the acquired data, the progress of the photoreaction is numerically simulated, and the resulting predicted conversion trace is subsequently compared to experiments that are performed with the 3D printed photoreactor.

In the following, obtained measurements on LED emission spectra, transmittance of glass vials, absorbance of relevant species, and importantly, the reaction quantum yield are discussed,

refer to Fig. 2. The structure of the numerical simulation is explained and the output of the algorithm, simulated light-attenuation maps as well as the resulting conversion trace is shown and compared to experimental outcomes, refer to Figs. 4 and 5. Furthermore, a related algorithm is introduced, which predicts the potential of the subsequently introduced novel λ-orthogonal ligation system employing two photoenol ligation reactions. The two simulation approaches address the challenges of accurately predicting the wavelength-dependent conversion and selectivity of photochemical experiments.

## Results

The numerical simulations perform all calculations with wavelength-dependent parameters. Emission spectra of LED 2 and 3 as well as the transmittance of the glass vials is shown in Fig. 2a. The initially obtained data of the spectra are represented by mathematical expressions, which can easily be processed in a numerical simulation. The transmittance gradually decreases from the visible wavelengths to about 315 nm, in line with previously reported values[22,23]. Further into the UV range, the transmittance dramatically decreases. For quantitative work, this cannot be ignored, and it was previously found that also quartz glass has a transmittance below 95%, despite being less dependent on the wavelength[24]. Furthermore, the emission spectrum of any LED shows that LEDs are not nearly monochromatic light sources, although the width of the emission spectra is usually <60 nm, while the full width at half maximum is in a range of 10–15 nm. Under the paradigm of precision photochemistry any wavelength dependence needs to be considered quantitatively, as even a minor part of a spectrum can have noticeable effects[23]. If an undesired overlap of emission spectrum and absorbance occurs in a complex photoreactive system, unwanted side products can be formed[25].

The absorbance of A, NEM, and AP was reported previously[21,26], but is shown in Fig. 2b to allow comparison with wavelength-dependent reaction quantum yields, obtained in this study from experiments that were carried out in triplicate, refer to Supplementary Information, section 2.2.1. As identified previously, the photoenol ligation using A is possible up to the wavelength of 420 nm[21]. Here, we report that the reaction quantum yield at this wavelength, $\Phi_{420\,nm,\,2.3\,mM} = 0.0026 \pm 0.0010$ is particularly low and increases with decreasing wavelength to reach a plateau in the range of 345–400 nm at $\Phi_{2.3\,mM} = 0.028 \pm 0.0037$. A further increase up to $\Phi_{307\,nm,\,2.3\,mM} = 0.115 \pm 0.023$ is observed with decreasing wavelength to 307 nm, whereas the reaction quantum yield appears to drop for wavelengths shorter than 307 nm. These findings can tentatively be interpreted by assigning various photochemical efficiencies to the transitions that are accessible by each wavelength. We proposed a similar interpretation previously for a related compound, an alkoxy-2-methylbenzaldehyde B, refer also to Fig. 3a[23]. While the assignment of an $n \rightarrow \pi^*$ and a $\pi \rightarrow \pi^*$ transition was straightforward with the alkoxy-derivative, the thioether-substituted compound appears to have less clearly separated transitions and the mixing between $n \rightarrow \pi^*$ and $\pi \rightarrow \pi^*$ states can occur.

Generally, a plateau in wavelength-dependent reaction quantum yields is in both cases an indication that the underlying transition is the main contributor to the absorption spectrum in the respective region and excitation into different vibrational levels within the same excited state quickly leads to the lowest vibrational level, respectively, before further photophysical processes occur. A slope in the plot of reaction quantum yields vs. wavelength, on the other hand, is either caused by two transitions competing for incident photons, or by a change in the efficiency of intersystem crossing to the triplet state dependent on the vibrational level that was reached

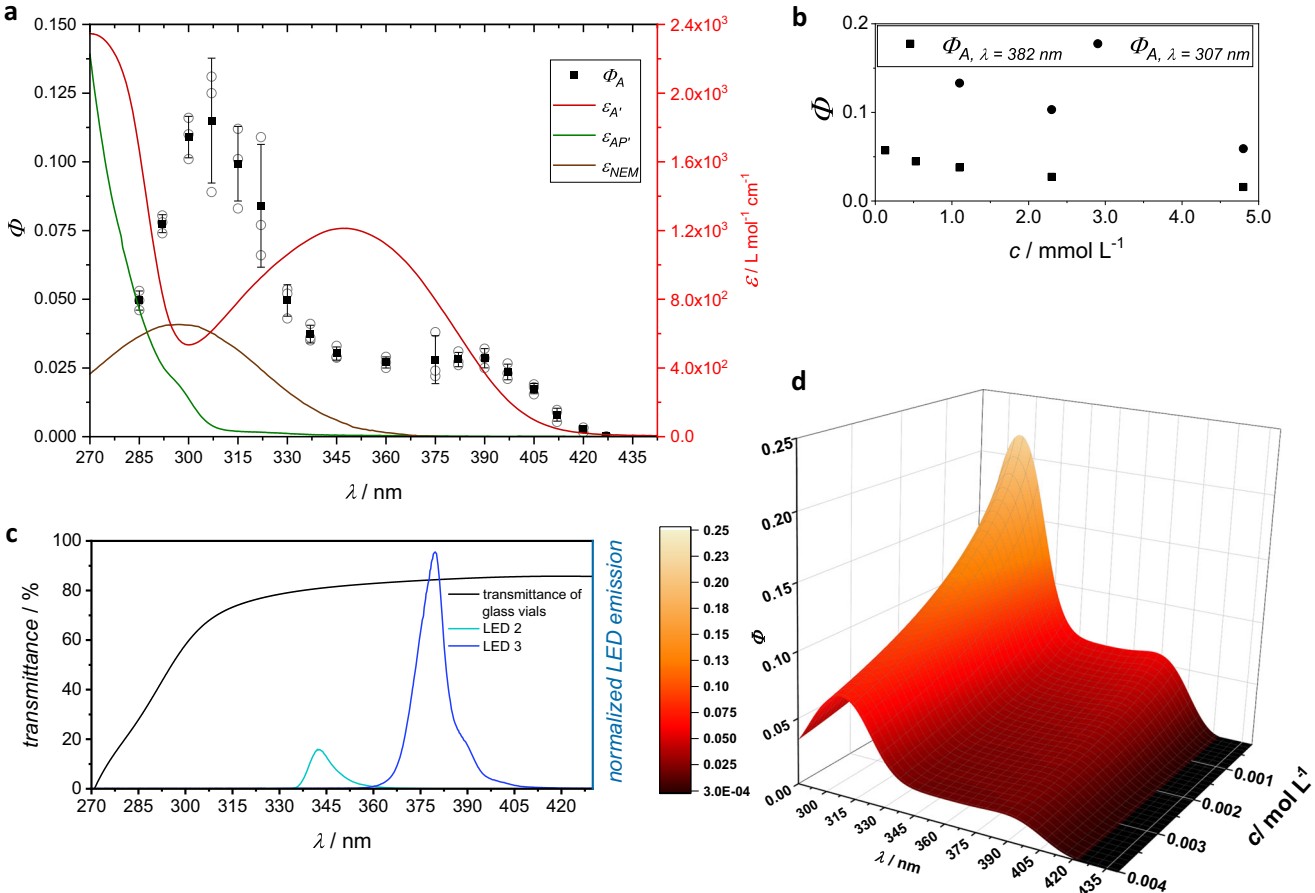

**Fig. 2 Wavelength- and concentration-dependent parameters. a** Molar attenuation coefficients of the chromophores as well as quantum yields as a function of the wavelength ($c_{A, initial} = 2.3$ mM). Reaction quantum yields were numerically calculated from the conversion of monochromatic tunable laser experiments, which were carried out in triplicate, refer to Supplementary Information, sections 1.5 and 2.2.1. The reaction quantum yields shown are averages of three experiments and the error bars show the standard deviation of these values. **b** Concentration-dependent reaction quantum yields ($\lambda =$ 382 and 307 nm). Single measurements. **c** Transmittance of the glass vials and emission spectra of LED 2 and 3. The emission spectra are normalized so that the integral represents the measured output power of the LED as obtained with the 3D printed LED detector scaffold, refer to Supplementary Information, sections 1.4 and 1.6. **d** Reaction quantum yield map based on mathematical expressions that are constructed to represent the obtained quantum yield values and extra/interpolated values, refer to Supplementary Information, section 2.3.

by the respective wavelength. In the former case, within each electronically excited state the reaction quantum yields may potentially be independent of the vibrationally excited level. Wavelength-dependent quantum yields have been reported for a range of light-induced reactions, but the underlying physicochemical reasons for such dependencies are not easily identified[27–29].

An equilibrium between two or more conformations of the aldehyde functionality and adjacent PEG chain in solution can explain the peak in quantum yields at 307 nm. The relevance of conformers becomes apparent through comparison of A with related compounds benzyl ether-substituted *o*-methylbenzaldehyde B (methyl 4-((2-formyl-3-methylphenoxy)methyl)benzoate) and dodecyl thioether-substituted *o*-methylbenzaldehyde C (2-(dodecylthio)-6-methylbenzaldehyde), refer to Fig. 3 and the Supplementary Information, section 2.4. The broad absorbance feature from 300 to 400 nm in the absorbance spectrum of A is not caused by a single electronic transition, but is rather a convoluted band, with several conformers of A contributing. The absorbance spectrum of C (Fig. 3a) shows that at least two absorbance bands contribute to the overall absorbance.

Similarly, the absorbance spectrum of A in less polar solvents, such as dichloromethane and tetrahydrofuran, refer to the Supplementary Information, section 2.4.1, confirm this observation.

A conformer, in which the aldehyde oxygen and thioether sulfur are close, as shown in Fig. 3b (O-S syn), indicates the presence of an intramolecular chalcogen bond, a noncovalent interaction well studied for small molecules with related structural motifs[30–32] as well as enzyme catalysis[33]. To gain a better understanding of relevant conformers, we carried out wavefunction and density functional theory calculations on the model molecule $C^{Me}$ to confirm that C or A are likely present in solution as conformers O-S syn, Ar-SR eclipsed, O-S anti, Ar-SR eclipsed and O-S anti, Ar-SR staggered, refer to Fig. 3b. In contrast, we conclude that B is only present in solution in significant amounts as the single conformer O-O anti eclipsed, explaining the comparably well-separated transitions, refer also to Figure S25. Each conformation can be expected to have characteristic electronic (vibronic) transitions, which are to a certain degree affected by the PEG chain in case of A. We have calculated relevant transition energies for the above-mentioned conformers of model compound $C^{Me}$, refer to Fig. 3a.

Our calculations place a $S_0 \rightarrow S_2$ transition of the O-S anti, Ar-SR staggered conformer with mixed $n\pi^*/\pi\pi^*$ character at 298 nm, refer also to the Supplementary Information, section 2.4.2. We propose that a respective conformer and transition in A is responsible for a significant fraction of absorbance between 285 and 345 nm and the maximum in the reaction quantum yield as

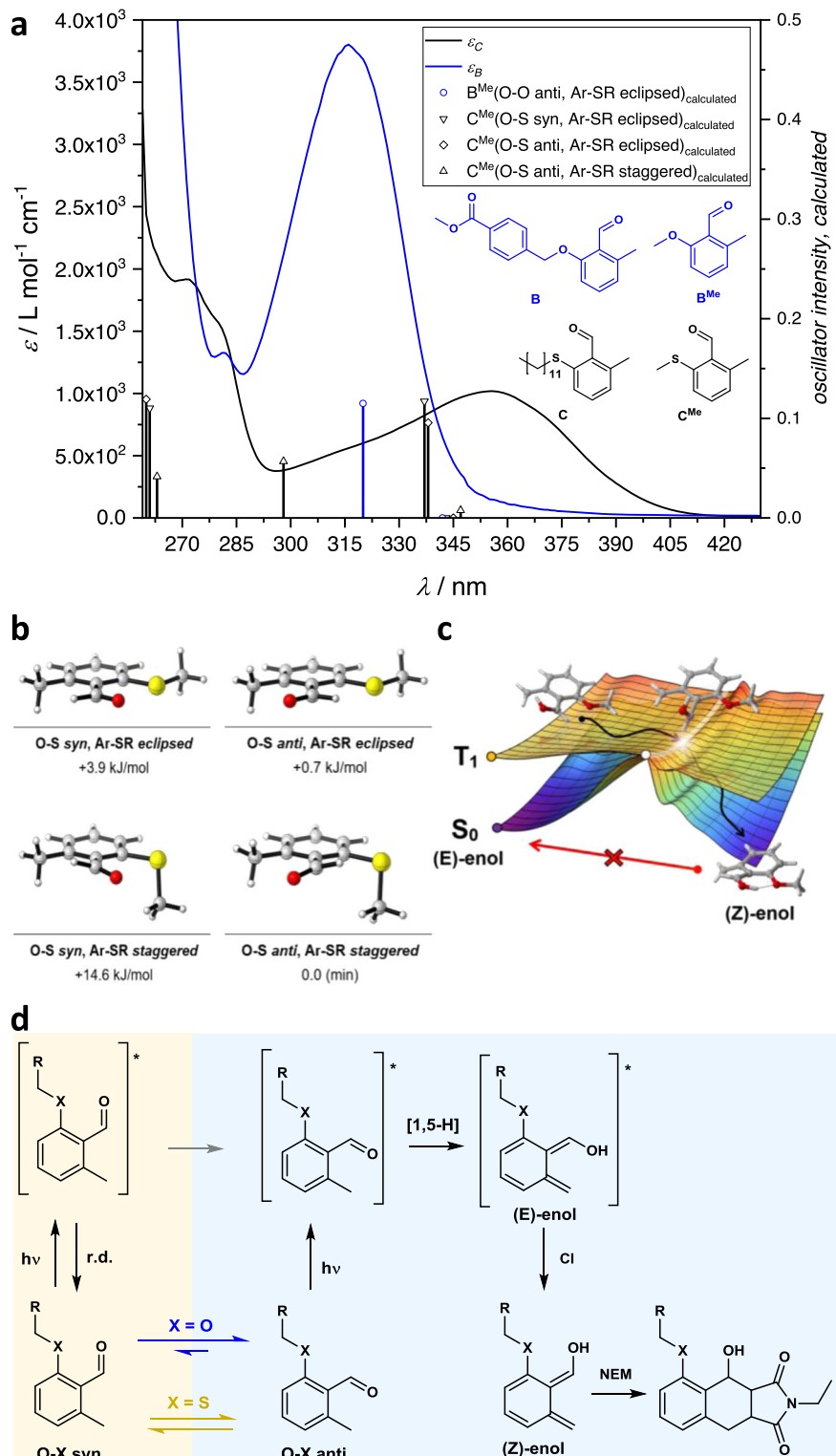

**Fig. 3 Aldehyde conformers affect wavelength-dependent quantum yields. a** Dependence of absorption on the substitution pattern of the *o*-methylbenzaldehyde. **b** Wavefunction and density functional theory calculations show a higher number of accessible conformations of the chromophore, which contribute to the observed absorption spectrum with both O-S syn and O-S anti conformers being possible conformations of A (C) in solution. For details regarding DFT-calculations, refer to the Supplementary Information, section 2.4.2. **c** Conical intersection seam enabling rapid ligation with high reaction quantum yields, as previously reported for the photoenol ligation of B. Adapted with permission from Elsevier[34]. **d** Proposed mechanism leading to reduced quantum yields of A in comparison to B, also causing the maximum in reaction quantum yield of A at 307 nm, due to a higher fractional contribution of the O-S syn conformer to the overall observed absorption of A at 307 nm (285–337 nm). For details, refer to the Supplementary Information, section 2.4.1.

determined by the fractional absorbance of this transition, refer also to Fig. 2a. The elevated reaction quantum yield for this conformer can be rationalized by the preferential conformation of the aldehyde, allowing a sigmatropic [1,5] H-shift to occur without prior bond rotation, refer to Fig. 3d. Furthermore, the following step (likely the passing of the initially formed (E)-enol through a conical intersection (CI) to form the ground state (Z)-enol, refer to Fig. 3c), which rapidly reacts with NEM) is an efficient pathway toward formation of the cycloadduct, as we found previously[23]. In contrast, we suggest that the O-S syn, Ar-SR eclipsed conformer increasingly contributes to absorbance at longer wavelengths than 307 nm, as a $S_0 \rightarrow S_2$ transition ($\pi\pi^*$ character) is expected at 337 nm with significant oscillator intensity, refer to Fig. 3b and the Supplementary Information, section 2.4.2. We expect a lower reaction quantum yield to be associated with this transition, as bond rotation in the excited state is required, before the [1,5] H-shift can occur.

The experimental investigation of thioether o-methylbenzaldehydes surprisingly revealed that the reaction quantum yield may be concentration-dependent, with higher reaction quantum yields at a lower initial concentration of the starting materials. We performed a detailed study, refer to Supplementary Information, section 2.2, subsection 2.2.2, to find the relation between the initial concentration and the observed reaction quantum yield. Due to the low sensitivity of $^1$H-NMR spectroscopy, high-resolution electrospray ionization mass spectrometry (ESI-MS) was employed for samples with low concentrations. The effect of potential ionization biases was reduced by evaluation of the double and triple charged (Na adducts) poly(ethylene glycol) bound species. A correction factor is determined, based on the measurement of samples with varied conversion of A to AP using both NMR and ESI-MS analysis. Indeed, for low initial concentrations of A, a reaction quantum yield of $\Phi_{382\,nm,\,0.13\,mM} = 0.057$ is found, whereas a gradual decrease toward $\Phi_{382\,nm,\,4.8\,mM} = 0.016$ is observed, see also the inset of Fig. 2b. The trend is qualitatively confirmed by $^1$H-NMR spectroscopy of samples with varied concentration at 307 nm. While a reaction quantum yield of $\Phi_{307\,nm,\,1.1\,mM} = 0.133$ is observed, it decreases to $\Phi_{307\,nm,\,4.8\,mM} = 0.059$ with increasing concentration. Possible reasons for the dependence of the reaction quantum yields on the initial concentration include self-quenching (e.g., triplet-triplet annihilation) at higher concentrations. For each dataset involving reaction quantum yields, a mathematical expression was formulated, which represents the observed trends and values of $\Phi$. The expressions were combined to an expression with two variables, which upon plotting lead to a map showing the extrapolated wavelength and concentration-dependent reaction quantum yields within the observed ranges, refer to Fig. 2c. The expression is included in the source code for the quantitative prediction of conversion, refer to Supplementary Information, section 2.4.

The algorithm presented in this work, a significant improvement of a previously introduced iterative method[26], is represented by the flow chart in Fig. S23, refer to Supplementary Information, section 2.5. Upon starting the program, information regarding the experiment that is to be simulated is requested. Combined with the absorbance spectra of relevant species, here A, NEM, and AP, all required data are collected, as the source code itself contains expressions for emission spectra, transmittance, and reaction quantum yields. The processes in the sample solution are simulated by calculation of light absorbance and reactivity within a defined number of segments, that the solution is divided into. The segments are defined by a stack of thin slices that the light sequentially penetrates.

After the prediction of the overall conversion within one second, the result is saved and subsequently a module in the code is redistributing the compounds across the segments, so that the solution is simulated to be fully mixed. This sequence is performed

until the entire irradiation time is simulated and the output is saved into an excel file. Importantly, within each of these iterative (temporal) steps, an entire iteration over the segments is performed for each wavelength increment. The distinction between the wavelength increments of 0.5 nm is important to reflect the wavelength dependence in all parameters from the emission spectrum, the transmittance, and absorbance of the species to the reaction quantum yields. The division of the solution into segments is particularly important for the code to be modular, as the complete mixing module can for future work be replaced by a simulation of stepwise mixing. Such a modification would be important, if a photoreaction is simulated, where the product of the reaction significantly absorbs at the wavelengths of irradiated light. The mixing of the solution in a real batch reactor is not ideal and depending on it a minor delay of the experimental reaction progress compared to simulated progress (with the assumption of ideal mixing) is anticipated. The conceptualization of a stepwise, incomplete mixing simulation, which is not based on arbitrary parameters exceeds the scope of this work and is the subject of future projects.

The predictions resulting from the photoreaction simulation with LED 2 and the respective experimental results are compared in Fig. 4. Predictions and experiments were carried out with and without the addition of the competitive absorber 2-hydroxy-5-nitrobenzaldehyde HNBA. For details, including the absorption spectrum of HNBA, refer to Supplementary Information, sections 1.7 and 2.7. The purpose of the addition of HNBA is to establish, whether the retardation effect of other light-absorbing molecules that otherwise do not interfere in the photoreaction can be predicted correctly. HNBA is treated in the simulation as a compound that absorbs light according to its molar attenuation coefficient, thereby affecting the overall fractional absorption of light in each simulated segment (second and wavelength) but it is not expected that any energy transfer from HNBA to A or vice versa occurs. In both cases, the observed conversion only slightly deviates from the predicted values, refer to Fig. 4a. Close examination of the predicted conversion trace shows that during the first 800 s of the reaction without HNBA addition, the predicted reaction rate (derivative of the plotted curve) increases slightly, before decreasing, refer to Fig. 4b. The increase in reaction rate is caused by the concentration-dependence of the reaction quantum yields. As the reaction progresses, A and NEM are consumed, leading to their concentrations to decrease—and thereby leading to increasing quantum yields over time. The rate of reaction significantly decreases, once light increasingly passes through the solution, as seen on the light-attenuation maps below (Fig. 4c) (right light-attenuation maps). This lower rate of conversion is not due to a change in the quantum yield, rather the effect of higher quantum yields at lower concentrations on the reaction rate is offset by the slowing of time-dependent conversion due to light passing through the solution without being absorbed.

Apart from delivering a prediction of the conversion trace of the experiment, the simulation simultaneously generates the data for the light-attenuation maps. Due to the segmented iterative approach, not only changes in (local) concentrations are simulated, but the spatiotemporal and wavelength-resolved residual light intensity during the experiment is calculated. For each second, segment, and wavelength, the incident number of photons is calculated for the respective step during simulation. The code is programmed to save the respective data in four cases: Before irradiation, after a third of the simulated time, after two-thirds, and finally upon completion of the algorithm. It can be concluded from the light-attenuation maps that initially the incident light is entirely absorbed, while after full conversion of A, the beam is for the best part passing through the solution. At this stage only a remaining minor amount of NEM causes some light to be absorbed.

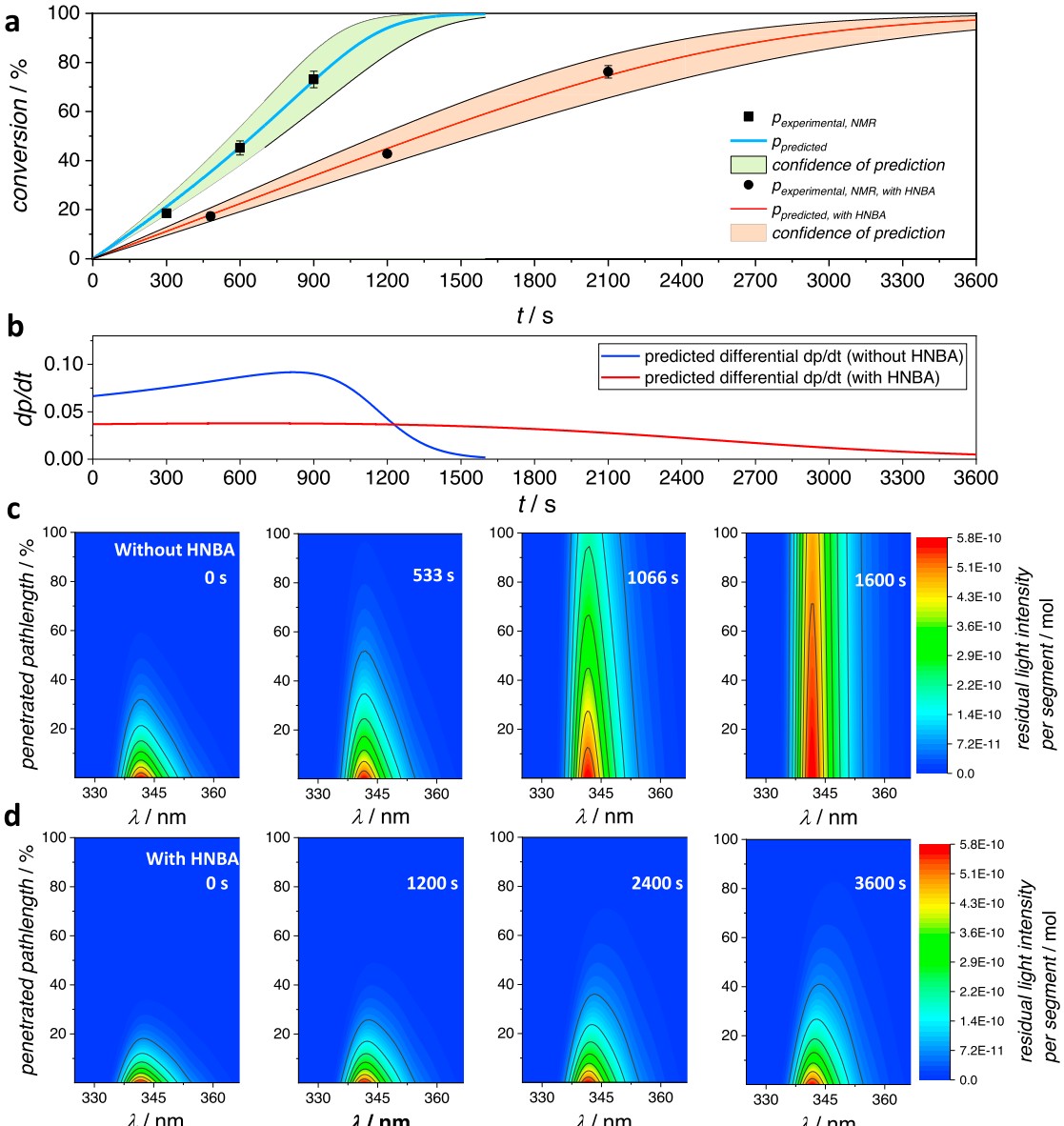

**Fig. 4 Predicted and observed conversion of LED irradiation experiments. a** Predicted (line) and experimental (dots, average of three replicates, error bar represents associated standard deviation) conversion of A and NEM to AP using the 3D-printed LED precision batch photoreactor using LED 2 ($\lambda_{max}$ = 343 nm). Predicted and observed conversion are compared for experiments with and without addition of 0.65 eq. 2-hydroxy-5-nitrobenzaldehyde HNBA. **b** Derivative of the predicted conversion plotted vs. irradiation time. **c** From left to right: Numerically calculated light-attenuation maps of the experiments without HNBA for selected irradiation times. Displayed is the number of photons (color-coded, unit: mol) within each wavelength range of 0.5 nm penetrating each individual segment for the duration of one second at the irradiation time as indicated in the top right of each light-attenuation map. **d** From left to right: Analogously calculated light-attenuation maps for the experiments with addition of HNBA.

In case of the addition of 0.65 eq. HNBA, the reaction is not only proceeding at a lower rate overall but also the rate of the reaction is more lowered as higher conversion is reached, because with increasing reaction time a higher percentage of photons is absorbed by HNBA rather than the reactant A, refer also to Fig. 4d. Such competitive absorption effects were recently qualitatively investigated and exploited for the design of chromatically orthogonal reactions[35,36]. In contrast, the results shown here impressively demonstrate that retardation effects through competitive absorption can now be predicted quantitatively. This visualization highlights that each photochemical experiment can be expected to exhibit its own specific kinetic behavior. The progress of the reaction would vary to some degree if any parameter were changed. A lower or higher amount of starting

materials not only implies a change in the reaction quantum yields that are applicable, but also leads to a changed number of photons being absorbed or passing through the solution over time. One can imagine endless examples where the dimensions of the reactor, the wavelength and intensity of incident light, amount of compounds, or addition of other light-absorbing compounds could alter the progress of the reaction. We propose that the most reliable prediction of photoreaction kinetics can only be achieved through numerical iterative simulation of the individual reaction progress using a reproducible photoreaction setup. A 3D-printed reactor is an example of how defined irradiation conditions can be established.

While a monochromatic, wavelength-tunable laser allows best control over irradiation, the use of an LED and 3D-printed

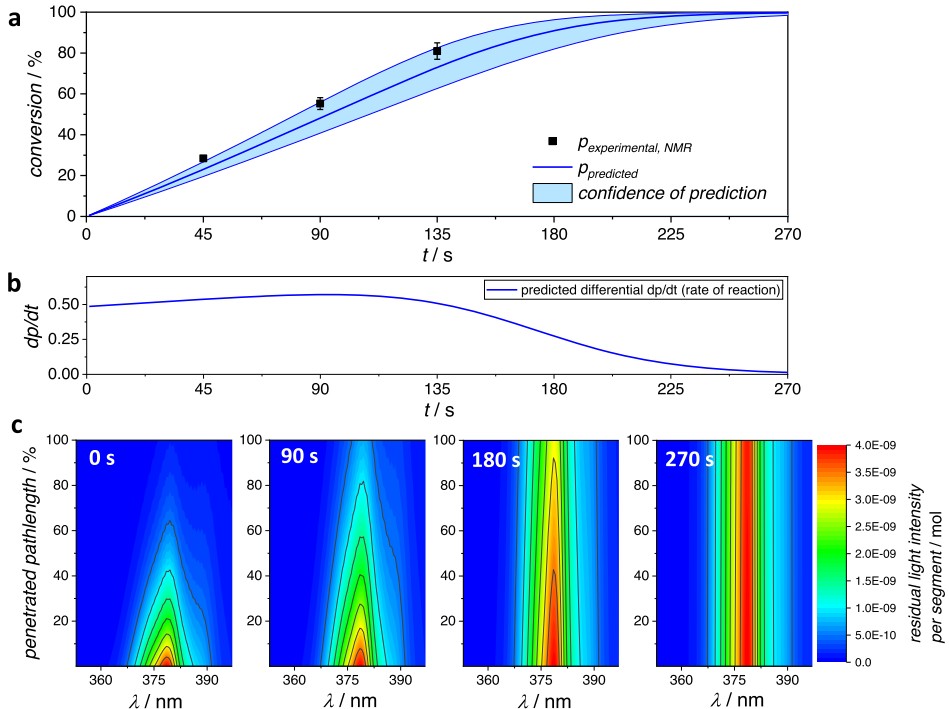

**Fig. 5 Predicted and observed conversion at varied wavelength. a** Predicted (line) and experimental (dots, average of three replicates, error bar represents associated standard deviation) conversion of A and NEM to AP using the 3D printed LED precision batch photoreactor using LED 3 ($\lambda_{max} =$ 380 nm). **b** Derivative of the predicted conversion vs. irradiation time. **c** From left to right: Numerically calculated light-attenuation maps of the same experiment for selected irradiation times. Displayed is the number of photons (color-coded, unit: mol) within each wavelength range of 0.5 nm penetrating each individual segment at the irradiation time as indicated in the top left of each light-attenuation map.

reactor with respective detector scaffold can provide an alternative pathway to estimate (apparent) quantum yields: The algorithm contains an option to replace the use of the quantum yield map with an arbitrary, fixed, apparent quantum yield, for details refer to the Supplementary Information, section 2.5. Thus, the conversion determined from an LED experiment can be used for the iterative determination of a quantum yield estimate, which is an approximation applicable to the wavelengths emitted by the respective LED.

To further verify the algorithm, we predicted the course of the reaction with LED 3, refer to Fig. 5. All calculations were conducted identically, whereas here the data associated with LED 3 is employed and the experiments are performed accordingly. In comparison to the previously shown experiment, a faster reaction is predicted and found here, partly caused by the stronger output of the LED, compare to the Supplementary Information, section 1.4. Similar to the results above, the derivative of the predicted curve initially slightly increases (for the first 90 s), before decreasing due to light increasingly passing through the solution instead of being absorbed, as can be traced with the light-attenuation maps (refer to Fig. 5b, c). Importantly, the experimental results closely mirror the predicted progress of the reaction. Further predictions and experiments involving LED 1 and LED 4 are shown in the Supplementary Information, section 2.8. In the case of LED 4, the emission spectrum overlaps with the absorption spectrum by only a very small degree. Yet, the predictive method provides a good conversion estimate. In case of LED 1, the prediction is correct only for small conversion values (20–30%), because with longer irradiation increasingly side products are being formed.

The precision of the predictive method for controlled photoreactions in a 3D printed LED batch photoreactor sets a new standard for photochemical experiments. Not only can monochromatic quantum

yields, obtained here with a sophisticated tunable laser setup, be applied in predicting experiments conducted in an inexpensive photoreaction setup, but the methods developed here are modular in nature and can be developed further.

Both determining wavelength-dependent reaction quantum yields and predicting photochemical kinetics helps considerably in the development of reaction systems that are controlled by the wavelength and/or intensity of light. Many combinations of chromophores, which can be addressed selectively by different wavelengths, are known[37–39]. A significant limitation of most examples is the competition between two reaction pathways, if one irradiation step activates both chromophores[40–42]. Selectivity can be achieved through control of the photo-kinetics by tuning both the degree of absorbance and the ratio of the relevant reaction quantum yields[26,43–45]. We have previously reported that a combination of A with a diaryltetrazole can be addressed with a very high degree of selectivity at two different wavelengths, employing LEDs centered at 285 and 380 nm[26]. The pro-fluorescent character of tetrazoles and the high absorbance of the product pyrazoline may be a disadvantage for certain applications, for example direct laser writing. The formation of light-absorbing cross-links in a photoresist makes the writing process challenging, as so-called micro-explosions are more likely[46]. In contrast, *o*-methylbenzaldehydes have been employed for direct laser writing in many occasions[47–50]. Considering the absorbance and quantum yields of A and of previously investigated alkoxy-substituted *o*-methylbenzaldehydes B[23], as shown in Fig. 6, it is not self-evident, whether a selective reaction of B in the presence of A can be achieved and which wavelength and which photon count may be suited best. A selective activation of A in the presence of B can be expected at wavelengths longer than 405 nm, as previously no detectable conversion was observed after irradiation of B at 405 nm with a defined photon count.

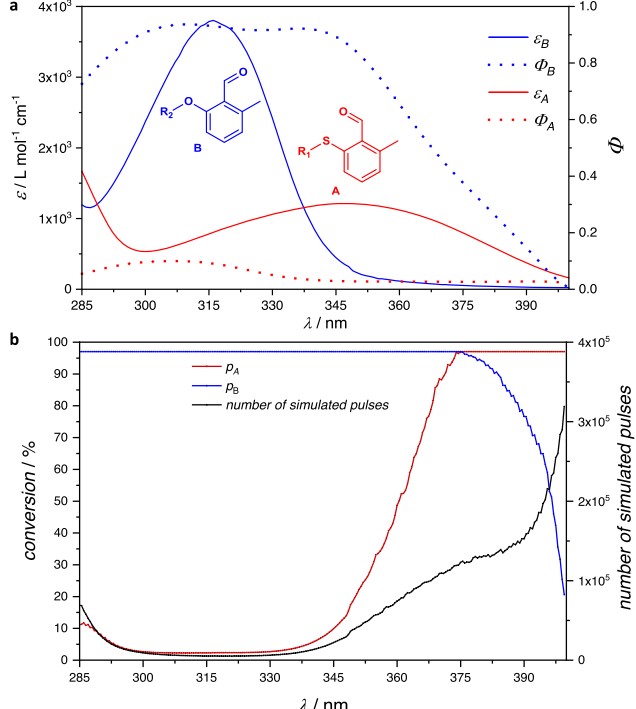

**Fig. 6 Predicted selectivity based on wavelength-dependent parameters. a** A comparison of wavelength-dependent molar attenuation coefficients (straight lines) and reaction quantum yields (dotted lines) shows that alkoxy-substituted *o*-methylbenzaldehyde B is both absorbing more light and exhibiting a higher reactivity[23] in the wavelength range of 289–337 nm than thioether-substituted *o*-methylbenzaldehyde A. **b** Numerical simulations predict an equally selective reaction outcome in the wavelength range of 300–330 nm for the irradiation of a mixture of 0.7 μmol A, 0.7 μmol B, and 1.65 μmol NEM in 0.11 mL CD₃CN.

Here, we present an algorithm capable of making wavelength-dependent predictions of the selectivity that can be expected for irradiation of a mixture of A, B, and NEM with monochromatic tunable laser light at varied wavelengths, refer to Supplementary Information, section 2.5. The output of the algorithm is the predicted conversion of both reactions of A or B with NEM at each wavelength, refer to Fig. 6. Each simulation is carried out with as many laser pulses as are required to reach a target conversion value for either of the two reactions. The conversion that is predicted for the other reaction, respectively, is a measure for the achieved selectivity, while the number of laser pulses is direct guidance for the experimental testing of the prediction. Surprisingly, despite the wavelength-dependent changes in the absorbance and the reaction quantum yields, the selectivity is predicted to be comparably good within the range of 300–330 nm. With these encouraging results, we tested the prediction experimentally, irradiating an equimolar solution of A and B with an excess of NEM in the presence of the internal standard TMB (1,3,5-trimethoxybenzene), refer to Fig. 7. Irradiation of a sample from the stock solution with the predicted number of monochromatic 325 nm tunable laser pulses at the respective laser energy (2.64 μmol total incident photons) leads to 93% conversion of B to BP, while A is mostly retained, with only 5% AP detectable. Despite the selectivity not being exactly as good as predicted, the outcome is still noteworthy and could be applied in situations where other chromophores are less suitable, such as in direct laser writing. Surprisingly, irradiation of the stock solution for 135 min at 415 nm (575 μmol total incident photons) led not only to 91% conversion of A to AP, but also caused 4% of B to be converted to BP. Yet, the reaction quantum yield of the here undesired reaction appears to be so low that even irradiation with this very high photon count only leads to a few percent conversion, allowing an overall comparably good selectivity. With an ever-increasing number of photoreactive molecules being synthesized, the possibilities to combine them in search of a specific, λ-orthogonal

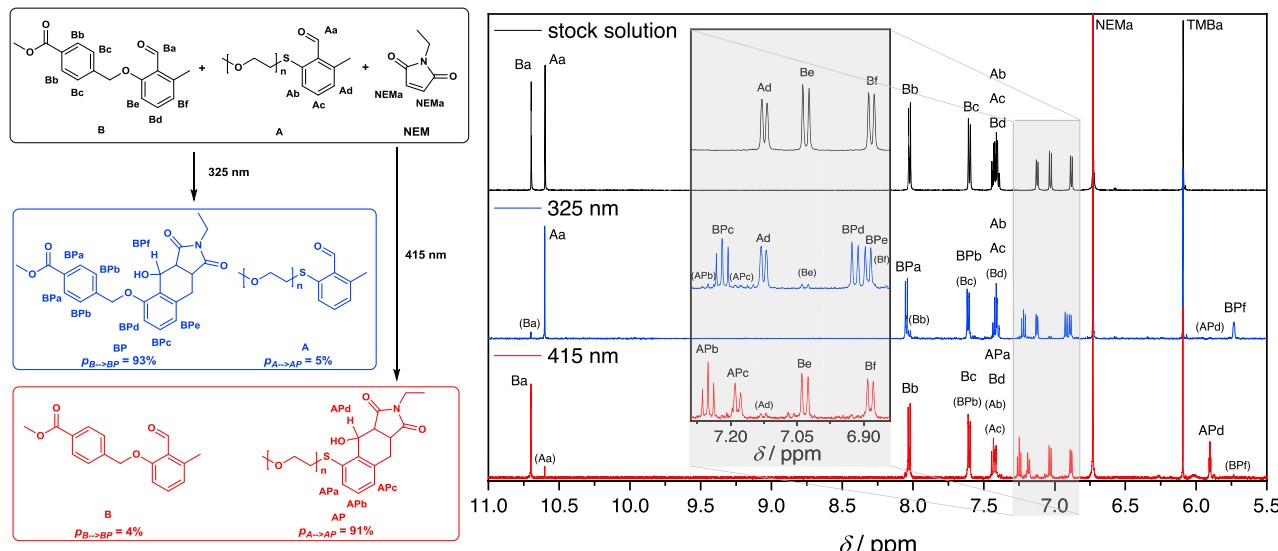

**Fig. 7 λ-orthogonal photoligation with *o*-methylbenzaldehydes.** A solution containing 0.7 mmol A, 0.7 mmol B, 1.65 mmol N-ethylmaleimide NEM, and 0.3 mmol internal standard trimethoxybenzene TMB (refer to the stock solution, top ¹H-NMR spectrum) is deoxygenated and irradiated with monochromatic tunable laser light at 325 nm (0.11 mL CD₃CN, 2.64 μmol total incident photons 184 mJ/pulse, 264 s irradiation, middle ¹H-NMR spectrum obtained after irradiation) and 415 nm (0.5 mL CD₃CN, 575 μmol total incident photons, 1.0 mJ/pulse, 135 min irradiation, bottom ¹H-NMR spectrum obtained after irradiation). The change in the integrals of the resonances of the aldehyde (Aa, Ba) and aromatic protons (Ad, Be, APb, APc, BPc, and BPd) as well as the proton in α-position to the newly formed hydroxy functionality (APd, BPe) all indicate that each wavelength leads to highly selective conversion. In both cases, only minor amounts of the undesired product are found.

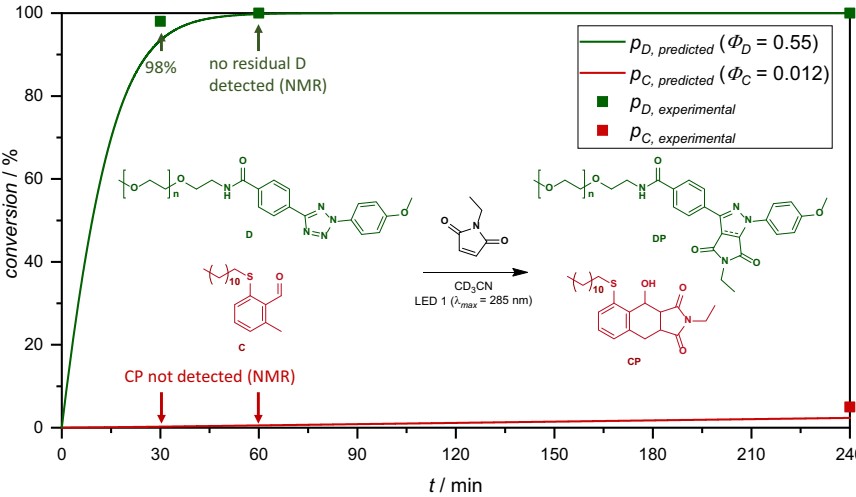

**Fig. 8 Predicted and observed λ-orthogonal photoligation.** Predicted and observed conversion (p) for the competing photoreactions of C and D with NEM during irradiation in acetonitrile with LED 1.

reaction system increase even faster: From $N$ photoreactive substrates, $N * (N − 1)/2$ different combinations can be created.

Comparison with the sequence-independent λ-orthogonal ligation system that we introduced earlier[26] and density functional theory calculations carried out within this study led us to question, whether the critical selectivity between tetrazole D (α-methyl-ω-4-(2-(4-methoxyphenyl)-*2H*-tetrazol-5-yl)benzamido poly(ethylene glycol)), refer to Fig. 8, and C under irradiation with LED 1 ($\lambda_{max} = 285$ nm) could be different to the previously reported selectivity between D and A. To this end, we determined reaction quantum yields of the conversion of C to CP as $\Phi_{C \to CP, 285\,nm} = 0.012 \pm 0.0017$ and $\Phi_{C \to CP, 360\,nm} = 0.008 \pm 0.0007$, each significantly lower than the observed reaction quantum yield of A, refer also to the Supplementary Information, section 2.9.2. Using the observed reaction quantum yield at 285 nm as well as the quantum yield estimate of $\Phi_{D \to DP, 285\,nm} > 0.55$[23,26] as apparent quantum yields, we modified the algorithm that predicts conversion of LED experiments to simulate the competing photoreaction between C and D with NEM, refer to Fig. 8. Experimentally, a faster than predicted conversion of tetrazole D is observed, with 98% conversion reached after 30 min of irradiation, while CP is undetectable both after 30 min and one hour of irradiation (C being retained quantitatively). Only after 4 h of irradiation is the undesired product CP detectable, underscoring the improved selectivity of C and D compared to A and D, despite competing absorbance and reactivity.

In contrast to the above-described combination of photoligation reactions, tetrazole D in combination with o-methylbenzaldehyde B leads to an unselective reaction outcome, as predicted and experimentally observed, refer to the Supplementary Information, section 2.9.3. Irradiation of D and B in presence of NEM and TMB was carried out at 313 nm, as the ratio of the molar attenuation coefficients of D and B at this wavelength (3.3) is the same as the one of B and A at 325 nm, compare to Fig. 6 and Fig. 7. The difference in the observed selectivity between the sets of experiments highlights that absorbance cannot be a sufficient guide for the design of selective photoreactions, but quantum yields in combination with a numerical simulation enable to predict whether good selectivity can be expected.

## Discussion

Our approach to use reaction quantum yields to assess which possible combination is promising as well as to predict the experimental parameters that are best suited to investigate the system is highly

effective. The computational tools presented in the current work are critical not only for the use of photons as the reactants of the 21st century but are the key to a smart and efficient design of sophisticated photochemically selective reaction systems. As the example in this work shows, the exchange of only an oxygen atom with a sulfur atom alters the absorbance and reactivity so dramatically that the photoreactions can be performed with high wavelength selectivity. This finding underpins the importance of the ongoing paradigm shift toward precision photochemistry, where both the in-depth characterization of reactive compounds and the rational experimental design and their computational treatment aid in the design of sophisticated applications.

Further to the prospect of identifying more sequence-independent λ-orthogonal reaction systems, methodologies to quantitatively predict photoreaction progress could be employed to identify the necessary light dose in bio-photochemical experiments. A major goal in photochemical research is developing substrates suitable for in vivo applications through red-shifting of the chromophore and enabling bioorthogonal photoligation or cleavage[51–53]. A predictive framework capable of correctly quantifying the effect of competitive absorption by biological tissue and resulting photo-kinetics could prove instrumental to design bio-photochemical applications. In addition, the question arises, whether the presented 3D-printed photoreactor could be applied to estimate quantum yields. Despite the determined reaction quantum yields not being monochromatic or as accurate as seen with other methods, it could offer guidance via fast and inexpensive experiments. If under the precision photochemistry paradigm, more data such as wavelength-dependent molar attenuation coefficients, LED emission spectra, glass transmittance values, LED output power values, and importantly quantum yields are reported, the literature forms a database containing all necessary information to predict more photochemical experiments quantitatively. When such an approach is combined with ab initio quantum chemical predictions of quantum yields[54] and absorbance spectra[55], one can finally envision the design of new complex applications based purely on hypothetical chemical structures and substitution patterns.

In summary, we show that the precision characterization of chromophores, including wavelength and concentration-dependent reaction quantum yields, combined with a well-defined 3D-printed irradiation setup and numerical simulations can be used to perform photochemical ligation in a highly reproducible, rationalized, and especially quantitatively predictable fashion employing polychromatic

light sources such as LEDs. The wavelength-resolved predictive method is also valid with the presence of competitively absorbing chromophores.

Furthermore, we introduce an algorithm, which can assess the potential wavelength-dependent selectivity of competing photoreactions and guide the respective experimental design. Such predictions enable us to identify an unprecedented $\lambda$-orthogonal ligation system based on two *o*-methylbenzaldehydes, which are either thioether or alkoxy-substituted. The change from an oxygen to a sulfur atom alters both the absorptivity and reactivity dramatically, allowing the selective transformation at two disparate wavelengths. The approach holds key potential to further rationalize photochemical reactivity. Challenges in the application of photochemical methods arising from the dependence of photochemical reaction progress on all reaction parameters can be overcome efficiently when the impact of any dependency is predictable. The computational prediction introduced here and what it can empower critically aids in the experimental design of photoreactions as well as enable sophisticated molecular control through light.

## Methods

**Materials**. The synthesis of *o*-methylbenzaldehydes employed here was reported before[21,26]. Other commercially available reactants and solvents, such as *N*-ethylmaleimide, 2-hydroxy-5-nitrobenzaldehyde, and CD$_3$CN were used as received.

**General routine of sample preparation for photoreactions**. Reactants and additives were weighed on an analytical balance, deuterated solvent was added with an Eppendorf pipette, before the stoichiometry of the stock solution was controlled $^1$H-NMR spectroscopically. Subsequently, appropriate parts of the stock solution were transferred to a 0.8 mL glass vial, a stir bar (3 mm) was added (in each case, except for the determination of quantum yields), the vial was crimped and the solution was deoxygenated by passing a stream of nitrogen gas for 10 min. Samples are always protected from ambient light, using aluminum foil and brown glass vials for storage of samples, otherwise sample preparation is as much as possible carried out in a fume-hood that is equipped with blue light-absorbing foil.

**Tunable laser photoreactions**. A pulsed, wavelength-tunable laser, including a Q-switched diode pumped Nd:YAG Laser, crystals for second and third harmonic generation, and an optical parametric oscillator among other optics was employed (repetition rate 20 Hz). Pulse energies (ranging from 70 μJ per pulse to 1 mJ per pulse) were measured with a Laser Power meter. For a full description, refer to Supplementary Information, sections 1.5 and 1.6.

**LED light-induced photoreactions**. Light-emitting diodes are mounted on a chip and an LED heatsink and connected to a variable power supply. The LED and LED heatsink are placed on a USB-powered fan on a magnetic stirrer (500 rpm). The 3D-printed photoreactor scaffold is placed on the LED/heatsink with the manual shutter closed. Magnetic stirrer and power supply are turned on (LED on), the sample is inserted into the scaffold. After 2 min, the manual shutter is opened and the timer for the irradiation time starts. After the desired irradiation time, the manual shutter is closed. Conversion is determined $^1$H-NMR spectroscopically. For details, refer to Supplementary Information, section 1.4.

**Characterization and instrumental analysis**. $^1$H-NMR spectroscopy: 600 MHz, 128 scans, *D*1: 2 s. The $\delta$-scale was referenced to the signal of residual solvent (CHD$_2$CN). Data were processed with MestReNova and OriginPro 9.1G. High-resolution electrospray ionization (orbitrap) mass spectrometry: The instrument (equipped with an HESI II probe) was calibrated in the *m/z* range 74–1822 using premixed calibration solutions (Thermo Scientific). A constant spray voltage of 4.7 kV and a dimensionless sheath gas of 5 were applied. The capillary temperature and the S-lens RF level were set to 320 °C and 62.0, respectively. The samples were dissolved with a concentration of 0.05 mg mL$^{-1}$ in a mixture of THF and MeOH (3:2) containing 100 mmol L$^{-1}$ of sodium trifluoroacetate and infused with a flow of 5 μL min$^{-1}$. Data were processed with XCalibur, mMass – Open Source Mass Spectrometry Tool, Microsoft Excel, OriginPro 9.1G, and custom-written algorithms (python source code).

**Numerical simulation**. Source code was written in the programming language python3, using Visual Studio Code for editing and executing code. The python modules and libraries 'math', 'openpyxl', and 'datetime' are used. For details, refer to Supplementary Information, sections 2.4 and 2.5.

**Density functional theory calculations**. All ab initio and density functional calculations were performed in ORCA 4.2.1[56]. Geometries and frequencies were

calculated at the B3LYP-D3BJ/Def2-TZVP level of theory[57,58]. Improved single-point energies were calculated using DLPNO-CCSD(T)/cc-pVTZ. For more details on the theoretical methodology, see Supplementary Information, section 2.4.2.

**Reporting summary**. Further information on research design is available in the Nature Research Reporting Summary linked to this article.

## Data availability
The data that support the findings of this study are available from the corresponding author upon reasonable request. Source data are provided with this paper.

## Code availability
Python source code is publicly available at https://github.com/jphmenzel/jpmphotochem. Source data are provided with this paper.

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

## Acknowledgements

We thank Florian Feist for providing the PEG-bound thioether o-methylbenzaldehyde A, Esa Jaatinen (QUT), and Sarah Walden (QUT) for assistance with LED output power measurements, Andreas-Neil Unterreiner (KIT) for helpful discussions, Irene Yarovsky (RMIT) for supporting B.B.N.'s participation in this project and Alexis Arriagada-Malone (NTech 3D printing) for assistance with 3D printing. This work was enabled by use of the Central Analytical Research Facility hosted by the Institute for Future Environments at QUT as well as the Additive Manufacturing Facility at QUT and NTech 3D printing. C.B.-K. acknowledges key continued support by the Queensland University of Technology (QUT) and the Australian Research Council (ARC) in the form of a Laureate Fellowship enabling his photochemical research program. C.B.-K. and J.P.B. acknowledge funding in the context of an Australian Research Council (ARC) Discovery project. C.B.-K. additionally acknowledges support by the Karlsruhe Institute of Technology in the context of the STN program of the Helmholtz association. J.P.M. acknowledges funding for his PhD studies by QUT.

## Author contributions

J.P.M. was responsible for the experimental conceptualization, writing of source code, running of simulations, conducting experiments, and writing of the original manuscript. B.B.N. carried out density functional theory calculations. C.B.-K. and J.P.B. motivated the study, supervised the project, discussed the data, edited the manuscript, and provided the conceptual framework for visible light photochemistry. All authors have given approval to the final version of the manuscript.

## Competing interests

The authors declare no competing interests.
