## [Peer Review File · Nature Communications]

REVIEWER COMMENTS

Reviewer #1 (Remarks to the Author):

Barner-Kowollick and co-workers present a very interesting and comprehensive manuscript which proposes a theoretical framework which quantitatively connecting through the consecutive processes in a photoreaction, i.e., 1) light irradiation, 2) penetration through media, 3) absorption by matter, 4) attenuation through the light path and 5) reaction of molecules in excited states. Only if the theoretical framework is qualitatively determined/interpreted can simulation/prediction of a photoreaction kinetics be possible. In this sense, the authors have set the first example in photochemistry to establish algorithms behind future possible automation, which is the highlight of this work and could justify its suitability for publication in Nature Communications. However, there are some parts which require some clarification. Please see below my suggestions and questions.

1. Referring to the quantum yield map (Figure 2b, black square, and Figure 2c), the authors correctly pointed out that the plateau within 360-400 nm in wavelength-dependent quantum yields resulted from relaxation of vibrational states of the same electronic state to its only ground vibrational state leading to the same photochemical transformation with a characteristic quantum yield. However, the slope/peak between 285-345 nm of molecule A was not correctly interpreted. The authors suggested two possibilities but neither of them seem plausible. Firstly, competition of excitation to different electronic states do not justify a surge in quantum yield at ca. 310 nm which is in the middle of two bright electronic states of molecule A; moreover, even if the 280 nm state is more reactive in the present reaction and can thus exert a synergistic effect with the 350 nm state, the quantum yield from 300 nm to 285 nm should go high rather than go low. Secondly, although it is possible that higher singlet electronic states can perform better in intersystem crossing to triplet states when energy levels and spin orbit coupling are suitable, again, there should not be a surge (i.e. 300 nm) in the middle of different states; furthermore, a small organic molecule composed of row 1-3 elements without any heavy atom, without large conjugation and without possible charge-transfer states, should not have notable intersystem crossing. In fact, 300 nm is not only the peak of the quantum yield, but also the peak of light absorption of the molecule NEM. This means that in the 300-330 nm region where the quantum yield is elevated, both reactants A and NEM are excited, and the reaction can occur between two excited molecules. Obviously, reaction between excited states are extremely active and can significantly promote the reaction quantum yield. The authors should better consider this perspective instead of other ambiguous or less likely explanations. This does not require "extensive theoretical studies" and a short discussion will be sufficient.

2. In the experiment where the authors determined the concentration-dependent quantum yields (Supplementary Information Table S10), the authors observed that with concentration going up, the apparent quantum yield went down. However, in the context of the authors' discussion, the quantum yield corresponds to the chemical activity of the reaction and thus should theoretically not be concentration-related. In fact, the quantum yield measured in the experiment is an apparent value which not only reflects the intrinsic reaction quantum yield, but also is affected by physical and photophysical behaviour of reactants. It is important to note that at higher concentrations, self-quenching of excited molecules tends to be more severe. Could the authors comment on the

likelihood that the concentration-dependent may be from self-quenching at high concentrations rather a change in the intrinsic quantum yield of the reaction? If this is the case, then it should not be considered a unique phenomenon/feature of thioether o-methylbenzaldehydes. Perhaps, the authors could consider to change quantum yield to apparent quantum yield.

3. The Beer-Lambert law may no longer be obeyed at high light intensities and/or low or high concentrations. This is due to the phenomenon of stimulated emission which means exposure of an excited molecule (excited by $h\nu$) to the same $h\nu$ irradiation will increase its possibility of de-excitation by emitting $h\nu$. As a result, saturation can be reached when identical population of the ground state and excited state is formed, which inhibits further absorption of photons. When saturation is reached, the beer-Lambert law lose effect as no more photons can be absorbed. Could the authors comment on whether light saturation should be considered in the theoretical framework and the algorithms and whether this may exist in the present work?

In conclusion, it is a very good and interesting paper which could fit well in Nat Comm after these clarifications.

Reviewer #2 (Remarks to the Author):

NCOMMS-20-25497 - Predicting Wavelength-Dependent Photochemical Reactivity and Selectivity
Menzel et.al.

Comments:

Menzel et al. report on the development of a method to numerically analyze and predict the outcome of photochemical reactions. Photochemical conversions currently are at the frontiers of many areas of chemical sciences and prediction and rationalization of light –induced transformations is usually very difficult. Methodology that help to predict and rationalize such multi-parameter processes is highly warranted and timely. The authors introduce methodology to quantitatively predict wavelength and time-dependent progress of a photochemical ligation based on experimental data and numerical simulation. In addition, competing photochemical reactions are studied focusing on an important challenge demonstrating selective orthogonal light-induced conversions. Specifically, the components of a photo-enolization based Diels-Alder model reaction, a benzaldehyde derivative and a maleimide, are used and analyzed regarding their absorbance and quantum yield at different wavelengths and concentrations employing a pulsed laser. From these results, a mathematical treatment results in a 3D map of the reaction. The so-obtained model allows to predict the kinetics and the outcome of the same reaction using widely used LEDs of different wavelength and intensity with high precision; the thoroughly parametrized algorithm allows for calculating the optimal conditions for achieving the λ -orthogonal ligation. The model is also able to consider the competing absorption by dyes added to the system, or to predict the optimal wavelength for orthogonal reactions with two competing aldehydes. There is a high demand for optimizing, characterizing and building reliable systems for better understanding of photochemistry in particular in complex biological environment. The described 3D printed LED batch photoreactor provides a simple system which can be adopted and implemented by other laboratories creating

reproducible and uniform data collection. In conclusion, the growing demand of reliable predictions for the outcome of photochemical reactions in a wide range of fields such as photobiology, photopharmacology, chemical biology, synthetic organic and inorganic chemistry makes this work of relevance for a broad readership of Nature Communications. This represents an important study and recommended for publication subject to the remarks, suggestions to improve some aspects and revisions indicated.

General remarks

- The authors could consider to restructure the introduction: discuss first the different aspects of a photochemical reaction (the two paragraphs about the laws of photochemistry) and then conclude the difficulty in design, transferability between setups and finally, the prediction of the kinetics. This would state the full extent of the challenge better and then conclude with their numerical approach as a step forward to solve these. However, this might reflect a personal preference.
- In order to predict the wavelength-dependent conversion and selectivity of the photochemical ligation, the authors chose intuitively an orthogonal (wavelength-selective) system. Even without the algorithm, it is apparent that only thioether-substituted o-methylbenzaldehyde (A) absorbs at wavelengths longer than 405 nm. On the other hand, alkoxy-substituted o-methylbenzaldehyde (B) has a significantly higher molar extinction coefficient in the UV region, making it more suitable for photo-induced ligation at these wavelengths. In order to emphasize the importance of the developed algorithm, it would be have been more appropriate to choose another two-component system with comparable absorptions in the UV region and less intuitive orthogonality/selectivity.
- Though the authors use a terminology which makes it easy to access the content of the work for scientists from diverse backgrounds, a series of inconsistencies when referring to the Supporting Information makes it difficult to follow the discussion fully. For instance
 - o Page 4 “Supplementary Information, section 2.3” It is section 2.2., subsection 2.2.2.
 - o Page 5 “flow chart in Figure S18, refer to Supplementary Information, section 2.5.”: F.S18 is not a flowchart but “Apparent mole fractions of AP in samples irradiated”. F.S18 is in section 2.2.; The flow chart the authors refer to is Figure S24 and can indeed be found in section 2.5.
 - o Page 7 “stronger output of the LED, compare to the Supplementary Information, section 1.5”. LEDs are in section 1.4. 1.5. is a laser setup.
- Moreover, the authors do not take full advantage of discussing their figures due to the lack of a sufficient number of cross-references. Especially, the conceptual figure 1 would benefit from a better connection to the relevant paragraphs in the introduction.
 - o The caption of figure 1 does not really describe Figure 1 but serves as an extension of the main text (a 3D printed LED photoreactor is discussed but a simplified Jablonski diagram is shown). The caption should be divided into two section describing the two panels shown. The abbreviations for the compounds should be explained in the caption. The sequence of the workflow shown on the left side of the figure should be indicated by an arrow or numbers. The meaning of “np” and “n” of the y-axis should be explained in the caption.
 - o The authors could consider extending figure 1 and display a scheme showing the aspects of the laws of photochemistry extensively discussed in the text.
 - o The readability of some figures is difficult. In particular, the dark blue in figure 1, the axis and legends in figure 2, the insert of Figure 2b (maybe make an extra sub-figure?) and the 3-letter code, the assignment for the proton signals is not clear, especially for BP and AP: BPf could be the OH signal or the neighboring C(C2OH)H signal. Please modify and clarify.
 - o Page 5 “In both cases, the observed conversion only slightly deviates from the predicted values”

refer here to the top part of Figure 3.

- Due to the clearly separated steps in the workflow, the authors could add topical subheadings as recommended by the author guidelines. Moreover, the main section “Discussion” reflects a summary and should hence not be titled a discussion. The discussion is provided with the main section “Results”.

Remarks to the Results and Methods

- Page 3: Wavelength-dependency of quantum yields: Please comment on the error of these values; in the SI in section 2.2.1. it is stated the error was estimated to be ca. 10%. It is not clear how this value was derived.
- Page 4: Concentration-dependency of quantum yields: Indeed the concentration dependency is somewhat unexpected. While the authors explain the wavelength-dependency on the possibility to access different transitions, the concentration-dependency is not further commented. Moreover, going to the SI (section 2.2.2), the quantum yields for the concentration dependency were assigned by a different method (mass spectrometry) than the wavelength-dependency. The calibration of the methods and the comparison to the NMR experiment are discussed earlier in the SI. However, as there is no comment in the main text or the SI, are the authors fully convinced by the data and interpretations provided that concentration-dependency is not due to a methodical error. Especially, since the deoxygenation procedure led to losses of reagents and consequently, varying concentrations (SI, 2.2.1.) Please elaborate this point.
- Page 5: “The rate of reaction significantly decreases, once light increasingly passes through the solution, as seen on the light attenuation maps below” I agree with the authors that in their model reaction, where the product absorbs more hypsochromically than the starting material, the solution becomes more transparent over time. However, I do not understand the conclusion the authors make here as they emphasize in the sentence before that the quantum yield increases over time. Please rephrase.
- Page 6: The addition of HNBA as additional dye competing for the photons is discussed. However, it is not clear which parameters are considered and whether the treatment is the same as the treatment of the reactants to make this prediction. A brief explanation would be helpful.
- Page 5 “the predicted reaction rate (derivative of the plotted curve) increases slightly, before decreasing” and Page 7 “Similar to the results above, the derivative of the predicted curve initially slightly increases (for the first 90 seconds)”: The authors depict the conversion in their figures but discuss the rate in the text. To make it clearer what is discussed a plot of the rate over time should be added.
- At the end of discussing the different LEDs on page 7, the authors comment briefly on LED 4 but not on LED 1. The SI reveals that the prediction of this LED1 is less good as for the other ones. The explanation in the SI is detailed but the authors should add a sentence or two to the main text and comment on this outcome as it nicely shows the limits of their method, which would contribute to the discussion.

Remarks to the SI

- Stacked NMR spectra over time: please enlarge the relevant areas and reduce the amount of baseline to visualize changes occurring. They are very difficult to see. The NMR spectrum in the main text is treated more reader-friendly, for instance.
- Section 2.1. “Apparent mole fractions are determined for each relevant polymer species with a

defined number of repeating units (refer to Figure S13)“ What is meant by polymer species?

- Figure S.14 and Table S8: How does the error from NMR experiments translate into the Mass? Why is the correction factor determined for each time point and not for the overall fit in S14? Would that not be more reliable and compensate for individual errors from degassing losses and pipetting (as I did not see that there was an internal standard used for Mass Spectrometry)?

- Losses due to degassing: In section S2.1. and section 2.2, it is stated that the reaction solutions were deoxygenated by nitrogen. In section 2.2.1., it is further stated that this led to losses of reagent (NEM).

- o Why the authors did not use, for instance, freeze-pump-thaw techniques to circumvent this problem?

- o As the quantum yield is sensitive to the concentration and probably not only NEM but also a part of the solvent was lost, can reproducibility of the results be assumed? How can you exclude that the concentration dependency of the quantum yield is not a methodically error?

- o Are the quantum yield measurements single measurements or were duplications or triplications done?

- Section 2.2.2. In line two, there is a cross reference to section 2.6. regarding mass spectrometry. Section 2.6. contains NMR data. Do you mean section 2.1.?

Minor comments

- First paragraph of the introduction: “This paradigm change is not only of academic interest, but has critical connotations for photo-pharmacology and biomedical applications”. I do not understand the differentiation between “academic interest” and “photopharmacology and biomedical applications”. I consider also the last two content of academic interest as well as synthetic photochemistry content of applied, i.e. industrial interest. Perhaps one should also explicitly mention the major developments currently in photo-redox catalysis.

- Page 9, first paragraph: “[...] not being exactly as good as predicted, the outcome is still noteworthy and could be applied in situations where other chromophores are unsuitable.” It is not clear which situations the authors refer to. The selectivity of the system is convincing and the amount of side product probably close to the error margin of the analysis.

- The end of the results: in vivo and ab initio do not need a hyphen

Reviewer #3 (Remarks to the Author):

Barner-Kowollik and coauthors report a detailed photophysical investigation of all parameters affecting a photocatalytic reaction. They use a self-printed reactor set up to fix the irradiation geometry for LED photoreactions and a tuneable laser set up. A model photoligation was investigated. In addition, computational tools were developed to predict the reaction outcome depending on changes in the reaction parameters, such as the irradiation wavelength.

The study is detailed, but in my opinion rather specialized. I have difficulties imagine a broader application of the method in photocatalysis, although better parameter determination and prediction of reaction outcomes would be highly desirable.

The determination of photophysical parameters in this study uses typical methods. These are available in a spectroscopy lab, but not always in organic synthetic laboratories. The 3-D printed reaction set up is slightly disappointing, as commercial alternatives have been reported with equal or

better performance (see for example the photoreactors by Merck and MacMillan, ICIQ start up, www.photoreactor.de). The prediction tool is interesting, but would only be useful for a broader audience if provided as tool with easy interface and input parameters a synthetic lab is able to provide.

Overall, this rather specialized study showcase that photochemical reactions can be understood and predicted in detail upon careful determination of all parameters. The manuscript is more suitable for a specialized journal focusing on photochemistry. It should be revised to be better accessible for potential users; currently the barrier for an synthetic organic chemist to use the methods is far too high.

Point-by-point response to reviewer's comments on manuscript NCOMMS-20-25497 *Predicting Wavelength-Dependent Photochemical Reactivity and Selectivity*

Reviewer #1:

Barner-Kowollik and co-workers present a very interesting and comprehensive manuscript which proposes a theoretical framework which quantitatively connecting through the consecutive processes in a photoreaction, i.e., 1) light irradiation, 2) penetration through media, 3) absorption by matter, 4) attenuation through the light path and 5) reaction of molecules in excited states. Only if the theoretical framework is qualitatively determined/interpreted can simulation/prediction of a photoreaction kinetics be possible. In this sense, the authors have set the first example in photochemistry to establish algorithms behind future possible automation, which is the highlight of this work and could justify its suitability for publication in Nature Communications. However, there are some parts which require some clarification. Please see below my suggestions and questions.

1. Referring to the quantum yield map (Figure 2b, black square, and Figure 2c), the authors correctly pointed out that the plateau within 360-400 nm in wavelength-dependent quantum yields resulted from relaxation of vibrational states of the same electronic state to its only ground vibrational state leading to the same photochemical transformation with a characteristic quantum yield. However, the slope/peak between 285-345 nm of molecule A was not correctly interpreted. The authors suggested two possibilities but neither of them seem plausible. Firstly, competition of excitation to different electronic states do not justify a surge in quantum yield at ca. 310 nm which is in the middle of two bright electronic states of molecule A; moreover, even if the 280 nm state is more reactive in the present reaction and can thus exert a synergistic effect with the 350 nm state, the quantum yield from 300 nm to 285 nm should go high rather than go low. Secondly, although it is possible that higher singlet electronic states can perform better in intersystem crossing to triplet states when energy levels and spin orbit coupling are suitable, again, there should not be a surge (i.e. 300 nm) in the middle of different states; furthermore, a small organic molecule composed of row 1-3 elements without any heavy atom, without large conjugation and without possible charge-transfer states, should not have notable intersystem crossing. In fact, 300 nm is not only the peak of the quantum yield, but also the peak of light absorption of the molecule NEM. This means that in the 300-330 nm region where the quantum yield is elevated, both reactants A and NEM are excited, and the reaction can occur between two excited molecules. Obviously, reaction between excited states are extremely active and can significantly promote the reaction quantum yield. The authors should better consider this perspective instead of other ambiguous or less likely explanations. This does not require "extensive theoretical studies" and a short discussion will be sufficient.

The reviewer queries our suggested interpretation of the wavelength-dependent quantum yields based on an assessment of the absorbance of the involved molecules and suggests replacing the suggested interpretation with another interpretation that the reviewer describes. We acknowledge that the origins of the wavelength-dependence, which indeed is unexpected and surprising, are highly interesting and relevant.

Firstly, to reassure the reviewer and the reader of the results that we found, we have now carried out more laser experiments (wavelength-dependent irradiation with the tunable laser, twice more for each wavelength between 285 nm and 420 nm). We determined the quantum yields for each experiment (from conversion determined by ¹H-NMR, in analogy to the already included experiments) by numerical simulation. The averages of the quantum yields (in triplicate) are shown in Figure 2b with the respective error bars. All trends that were described are confirmed and the values in the UV range are associated with an error of close to 12%, while the error over the entire range is 15%. The error reported here is comparable to similar experiments, carried out in a previous study (Menzel, J. P. *et al.* Wavelength Dependence of Light-Induced Cycloadditions. *J. Am. Chem. Soc.* **139**, 15812-15820, (2017).).

a university for the real world[®]

Secondly, we have investigated the absorbance of a different derivative of compound A (a dodecyl substituted thioether C rather than a PEG-bound thioether A) as well as solvent-dependent absorbance of both derivatives. We found that the broad absorbance band from 300 nm to 420 nm is not caused by a single transition, but rather at least two underlying transitions, which can be seen more clearly in different solvents (tetrahydrofuran, dichloromethane, because these are less polar solvents than acetonitrile) or with the differently substituted compound C, refer to the Supplementary Information, section 2.4.1.

Consequently, we developed and explain in detail a coherent hypothesis for the experimentally observed behaviour in the revised version. This hypothesis is outlined in the manuscript and explained in detail in the Supplementary Information, section 2.4. Supporting evidence for our explanation is found by comparing the substrate A to derivatives of it. Both the absorbance of related compounds and underpinning theoretical calculations (wavefunction and density

functional theory) carried out by Benjamin Noble support our now comprehensive explanation. For details, please refer to the manuscript and Supplementary Information.

The suggestion that N-ethylmaleimide is excited and then two excited states react particularly efficiently with each other is contrary to the mechanism that is established for this ligation reaction. According to the Woodward-Hoffmann rules, the Diels-Alder [4+2] cycloaddition of *o*-quinodimethanes (dienes, which can also thermally be generated) with electron poor alkenes (such as NEM) is thermally allowed. The *o*-quinodimethane can further be expected to be present in the ground state, rather than an excited state, due to the possibility of a conical intersection connecting the triplet and ground state surface of the *o*-quinodimethane. Further, the energetic barrier for this specific cycloaddition can be anticipated to be very low already, as was calculated for a related *o*-quinodimethane (with an ether instead of a thioether, Wavelength Dependence of Light-Induced Cycloadditions, *Journal of the American Chemical Society*, **2017**, 139, 44, 15812-15820), making this step a rapid, 'non-efficiency-determining' step.

2. In the experiment where the authors determined the concentration-dependent quantum yields (Supplementary Information Table S10), the authors observed that with concentration going up, the apparent quantum yield went down. However, in the context of the authors' discussion, the quantum yield corresponds to the chemical activity of the reaction and thus should theoretically not be concentration related. In fact, the quantum yield measured in the experiment is an apparent value which not only reflects the intrinsic reaction quantum yield, but also is affected by physical and photophysical behaviour of reactants. It is important to note that at higher concentrations, self-quenching of excited molecules tends to be more severe. Could the authors comment on the likelihood that the concentration-dependent may be from self-quenching at high concentrations rather a change in the intrinsic quantum yield of the reaction? If this is the case, then it should not be considered a unique phenomenon/feature of thioether *o*-methylbenzaldehydes. Perhaps, the authors could consider to change quantum yield to apparent quantum yield.

We thank the reviewer for their helpful comment. We have changed the description of the quantum yield to 'reaction quantum yield' as this might point more clearly towards the fact that not a quantum yield of a single photophysical step is described, but rather the quantum yield of the entire reaction, spanning mechanistically from absorbance of a photon to formation of the final product. In the literature, there exist numerous reports of quantum yields depending on parameters such as the irradiation wavelength, solvent or even proximity of the chromophores to each other. It may be important to conclude here that for some reactions an 'intrinsic' quantum yield can never be established for the reactant structure, as the reactivity depends on the environment. In contrast, other reactions may not show any noticeable dependence on such parameters and a quantum yield can be assigned universally to the chromophore structure. As we report the reaction quantum yield, the number of formed product (cycloadducts) molecules per photon absorbed by the photoreactive chromophore, this reaction quantum yield may inherently be susceptible to self-quenching, if this is a process that occurs under the respective conditions. We have included a note in the manuscript to point out the possibility of self-quenching as an explanation of the dependence of the reaction quantum yield on the initial concentration.

3. The Beer-Lambert law may no longer be obeyed at high light intensities and/or low or high concentrations. This is due to the phenomenon of stimulated emission which means exposure of an excited molecule (excited by $h\nu$) to the same $h\nu$ irradiation will increase its possibility of de-excitation by emitting $h\nu$. As a result, saturation can be reached when identical population of the ground state and excited state is formed, which inhibits further absorption of photons. When saturation is reached, the Beer-Lambert law lose effect as no more photons can be absorbed. Could the authors comment on whether light saturation should be considered in the theoretical framework and the algorithms and whether this may exist in the present work?

In the laser experiments included in our study, the photon numbers of each individual laser pulse (monochromatic nanosecond laser pulse) are in the range of 0.1 to 0.9 nmol, whereas the (initial) amount of compound **A** is 500 nmol for each sample (study of wavelength-dependence). Stimulated emission can only occur in these experiments if a photon of equal wavelength (monochromatic light source) is absorbed by a molecule that has already formed an excited state within the duration of the same laser pulse. The laser pulse frequency is 20 Hz, which leaves 50 ms between each laser pulse, which is sufficient time for an initially formed excited state to undergo relaxation processes and not be susceptible to STED processes with the same wavelength. Thus, saturation cannot be reached and STED processes can be disregarded. In the case of LED experiments, one does not expect STED processes at all for similar reasons, particularly the number of photons delivered per nanosecond is much lower.

On the other hand, the formation of potentially sufficiently long living intermediate states with different absorbance than reactant or product can, if relevant in the experiment, lead to a slightly different absorption behaviour in the real experiment compared to the simulated experiment. Yet, as mentioned above, the number of excited states formed during one laser pulse is very small compared to the initial amount of starting materials. The assumption of intermediate state absorption being negligible is made, also because the intermediate states cannot be isolated and their lifetime and

CRICOS No. 00213J

absorbance spectrum cannot be measured easily. Such investigations can be the subject of future investigations, but the applicability and predictive power of the algorithm show that such effects are not relevant here and can for the purpose of photochemical simulation indeed be disregarded in the case shown.

In conclusion, it is a very good and interesting paper which could fit well in Nat Comm after these clarifications.

Reviewer #2:

Menzel et al. report on the development of a method to numerically analyze and predict the outcome of photochemical reactions. Photochemical conversions currently are at the frontiers of many areas of chemical sciences and prediction and rationalization of light –induced transformations is usually very difficult. Methodology that help to predict and rationalize such multi-parameter processes is highly warranted and timely. The authors introduce methodology to quantitatively predict wavelength and time-dependent progress of a photochemical ligation based on experimental data and numerical simulation. In addition, competing photochemical reactions are studied focusing on an important challenge demonstrating selective orthogonal light-induced conversions.

Specifically, the components of a photo-enolization based Diels-Alder model reaction, a benzaldehyde derivative and a maleimide, are used and analyzed regarding their absorbance and quantum yield at different wavelengths and concentrations employing a pulsed laser. From these results, a mathematical treatment results in a 3D map of the reaction. The so-obtained model allows to predict the kinetics and the outcome of the same reaction using widely used LEDs of different wavelength and intensity with high precision; the thoroughly parametrized algorithm allows for calculating the optimal conditions for achieving the λ -orthogonal ligation. The model is also able to consider the competing absorption by dyes added to the system, or to predict the optimal wavelength for orthogonal reactions with two competing aldehydes. There is a high demand for optimizing, characterizing and building reliable systems for better understanding of photochemistry in particular in complex biological environment. The described 3D printed LED batch photoreactor provides a simple system which can be adopted and implemented by other laboratories creating reproducible and uniform data collection. In conclusion, the growing demand of reliable predictions for the outcome of photochemical reactions in a wide range of fields such as photobiology, photopharmacology, chemical biology, synthetic organic and inorganic chemistry makes this work of relevance for a broad readership of Nature Communications. This represents an important study and recommended for publication subject to the remarks, suggestions to improve some aspects and revisions indicated.

We thank the reviewer for their positive assessment of our work.

General remarks

The authors could consider to restructure the introduction: discuss first the different aspects of a photochemical reaction (the two paragraphs about the laws of photochemistry) and then conclude the difficulty in design, transferability between setups and finally, the prediction of the kinetics. This would state the full extent of the challenge better and then conclude with their numerical approach as a step forward to solve these. However, this might reflect a personal preference.

We thank the reviewer for their comment. However, also in the light of the other reviewers' comments, we have on this occasion opted for leaving the introduction unchanged.

In order to predict the wavelength-dependent conversion and selectivity of the photochemical ligation, the authors chose intuitively an orthogonal (wavelength-selective) system. Even without the algorithm, it is apparent that only thioether-substituted o-methylbenzaldehyde (A) absorbs at wavelengths longer than 405 nm. On the other hand, alkoxy-substituted o-methylbenzaldehyde (B) has a significantly higher molar extinction coefficient in the UV region, making it more suitable for photo-induced ligation at these wavelengths. In order to emphasize the importance of the developed algorithm, it would be have been more appropriate to choose another two-component system with comparable absorptions in the UV region and less intuitive orthogonality/selectivity.

We thank the reviewer for pointing this out, noting that with the reactions investigated, the reader may be inclined to think that the selectivity could be predicted based purely on the absorbance of the chromophores. We have taken the reviewer's advice and investigated a different combination of molecules with less intuitive selectivity, see the Supplementary Information, section 2.9.3. Particularly, we chose to select molecules, for which the quantum yield is known, but we also chose a wavelength, for which the ration of molar attenuation coefficients is equal to the one of the

CRICOS No. 00213J

reaction combination already included in the paper. Thus, any different predicted and observed selectivity for the newly included experiments highlights the importance of the knowledge of quantum yields and importantly, the predictive algorithm.

We predicted the wavelength-dependent conversion of tetrazole D and *o*-methylbenzaldehyde B in presence of NEM and TMB. We then irradiated three samples drawn from the same stock solution with each 1200 laser pulses (313 nm) and determined conversion for each reaction channel. We observed a small deviation between the predicted and observed selectivity, which shows the potential and the limitations of the predictive method better. The deviation can in fact be explained with the quantum yield of the tetrazole being an estimate ($\Phi_D > 0.55$).

Though the authors use a terminology which makes it easy to access the content of the work for scientists from diverse backgrounds, a series of inconsistencies when referring to the Supporting Information makes it difficult to follow the discussion fully. For instance

Page 4 “Supplementary Information, section 2.3” It is section 2.2., subsection 2.2.2.

This is corrected now.

Page 5 “flow chart in Figure S18, refer to Supplementary Information, section 2.5.”: F.S18 is not a flowchart but “Apparent mole fractions of AP in samples irradiated”. F.S18 is in section 2.2.; The flow chart the authors refer to is Figure S24 and can indeed be found in section 2.5.

Page 7 “stronger output of the LED, compare to the Supplementary Information, section 1.5”. LEDs are in section 1.4. 1.5. is a laser setup.

We thank the reviewer for pointing out these mistakes, which are corrected now.

Moreover, the authors do not take full advantage of discussing their figures due to the lack of a sufficient number of cross-references. Especially, the conceptual figure 1 would benefit from a better connection to the relevant paragraphs in the introduction.

We added cross-references to embed the figures better into the text.

The caption of figure 1 does not really describe Figure 1 but serves as an extension of the main text (a 3D printed LED photoreactor is discussed but a simplified Jablonski diagram is shown). The caption should be divided into two sections describing the two panels shown. The abbreviations for the compounds should be explained in the caption. The sequence of the workflow shown on the left side of the figure should be indicated by an arrow or numbers. The meaning of “np” and “n” of the y-axis should be explained in the caption.

We thank the reviewer for the suggestion, this has been incorporated now.

The authors could consider extending figure 1 and display a scheme showing the aspects of the laws of photochemistry extensively discussed in the text.

We added a panel to Figure 1, containing each a visualization or short summary of each law relevant to precision photochemistry with LEDs.

The readability of some figures is difficult. In particular, the dark blue in figure 1, the axis and legends in figure 2, the insert of Figure 2b (maybe make an extra sub-figure?) and the 3-letter code, the assignment for the proton signals is not clear, especially for BP and AP: Bp could be the OH signal or the neighboring C(C2OH)H signal. Please modify and clarify.

We thank the reviewer for their helpful suggestions and amended the figures accordingly.

Page 5 “In both cases, the observed conversion only slightly deviates from the predicted values” refer here to the top part of Figure 3.

This is now linked to the respective Figure (now Figure 4).

Due to the clearly separated steps in the workflow, the authors could add topical subheadings as recommended by the author guidelines. Moreover, the main section “Discussion” reflects a summary and should hence not be titled a discussion. The discussion is provided with the main section “Results”.

We have modified the respective titles and added further subheadings for clarity.

Remarks to the Results and Methods

Page 3: Wavelength-dependency of quantum yields: Please comment on the error of these values; in the SI in section 2.2.1. it is stated the error was estimated to be ca. 10%. It is not clear how this value was derived.

The error was assumed to be an appropriate estimate based on similar experiments that we previously carried out (Wavelength Dependence of Light-Induced Cycloadditions, *Journal of the American Chemical Society*, 2017, 139, 44, 15812-15820).

We have now carried out more experiments (two more laser irradiation experiments for each wavelength within the range of 285 nm to 420 nm). The quantum yields are thus determined from experiments carried out in triplicate and the error is included in the manuscript (error bars in Figure 2b). Within the UV range, the average error is close to 12%, while over the entire range the error is 15%. These values are in relatively good agreement with our initial estimation. Importantly, the fitted curve does not change significantly taking into account the additional experiments and all trends that were observed are confirmed.

Page 4: Concentration-dependency of quantum yields: Indeed the concentration dependency is somewhat unexpected. While the authors explain the wavelength-dependency on the possibility to access different transitions, the concentration-dependency is not further commented. Moreover, going to the SI (section 2.2.2), the quantum yields for the concentration dependency were assigned by a different method (mass spectrometry) than the wavelength-dependency. The calibration of the methods and the comparison to the NMR experiment are discussed earlier in the SI. However, as there is no comment in the main text or the SI, are the authors fully convinced by the data and interpretations provided that concentration-dependency is not due to a methodical error. Especially, since the deoxygenation procedure led to losses of reagents and consequently, varying concentrations (SI, 2.2.1.) Please elaborate this point.

We have corrected the concentrations according to the losses of solvent that we observed in the deoxygenation procedure with varying solvent volumes. Further, we have assessed the concentration dependence at 307 nm using NMR spectroscopy and amended the manuscript and Supplementary Information accordingly. The results clearly support all previously found trends. Thus, all results can be trusted and no methodical errors are present anymore. The loss of a small amount of solvent led to slightly changed concentrations, but this did not change the observed trend. The concentration dependence may arise from self-quenching, which can be expected to be more relevant at higher concentrations. The quantum yields reported are reaction quantum yields, which means that quenching processes may affect these values: The reaction quantum yield is defined as the number of product molecules formed divided by the number of photons absorbed by the reactive chromophore.

Page 5: “The rate of reaction significantly decreases, once light increasingly passes through the solution, as seen on the light attenuation maps below” I agree with the authors that in their model reaction, where the product absorbs more hypsochromically than the starting material, the solution becomes more transparent over time. However, I do not understand the conclusion the authors make here as they emphasize in the sentence before that the quantum yield increases over time. Please rephrase.

We have added a sentence that should clarify our statements regarding the quantum yields and predicted rate of the reaction. With photochemical experiments being highly constrained by the irradiation geometry, the expected time-dependent conversion is affected, when the solution becomes transparent.

A thought experiment may highlight this further: Consider the expected kinetics of the reaction with the same amounts of compounds, concentrations and photon numbers, but varied area and depth of the solution and area of radiation. If the same amount of photons is spread over 1 square meter and the solution too, the solution is a very thin layer (as the volume is equal). In case of such a thin layer, many photons would pass through the solution, as they are not absorbed and the rate of reaction would slow down much earlier. Considering the opposite case, a radiation beam and solution inside what would be a long, thin glass tube with 1 square mm in diameter, photons would likely almost all be absorbed during the reaction right until almost all the reactant is used up. It becomes obvious that geometrical parameters can strongly affect reaction kinetics in photochemistry.

Page 6: The addition of HNBA as additional dye competing for the photons is discussed. However, it is not clear which parameters are considered and whether the treatment is the same as the treatment of the reactants to make this prediction. A brief explanation would be helpful.

We have added an explanation that should clarify this question.

Page 5 “the predicted reaction rate (derivative of the plotted curve) increases slightly, before decreasing” and Page 7 “Similar to the results above, the derivative of the predicted curve initially slightly increases (for the

CRICOS No. 00213J

first 90 seconds)”: The authors depict the conversion in their figures but discuss the rate in the text. To make it clearer what is discussed a plot of the rate over time should be added.

We thank the reviewer for this suggestion, and we have included the respective plots now in the respective Figures.

At the end of discussing the different LEDs on page 7, the authors comment briefly on LED 4 but not on LED 1. The SI reveals that the prediction of this LED1 is less good as for the other ones. The explanation in the SI is detailed but the authors should add a sentence or two to the main text and comment on this outcome as it nicely shows the limits of their method, which would contribute to the discussion.

We agree that this is a suitable note to make in the main paper and we have added it accordingly.

Remarks to the SI

Stacked NMR spectra over time: please enlarge the relevant areas and reduce the amount of baseline to visualize changes occurring. They are very difficult to see. The NMR spectrum in the main text is treated more reader-friendly, for instance.

We thank the reviewer for this comment. We have made the changes accordingly.

Section 2.1. “Apparent mole fractions are determined for each relevant polymer species with a defined number of repeating units (refer to Figure S13)” What is meant by polymer species?

We have changed this to macromolecular species, as it refers to the individual macromolecules and we have added a brief explanation to clarify. Mass spectrometry discriminates between the molecules depending on the number of repeating units in the polymer backbone (polymer distribution). To confirm that there is no significant bias introduced by choosing on peak with an arbitrary number of repeating units, our analysis considers the entire polymer distribution. For the samples that we investigated here, we find that the apparent mole fraction is independent of the number of repeating units, which is in line to our previous reports of this analysis method.

Figure S.14 and Table S8: How does the error from NMR experiments translate into the Mass? Why is the correction factor determined for each time point and not for the overall fit in S14? Would that not be more reliable and compensate for individual errors from degassing losses and pipetting (as I did not see that there was an internal standard used for Mass Spectrometry)?

Due to the method of using NMR results to ‘calibrate’ the output of the mass spectra, the mass spectrometric result cannot be more accurate than an NMR measurement, despite the better sensitivity of mass spectrometry (enabling results at low compound concentrations / amounts of substrate). We estimate the error of a quantum yield that is determined from a mass spectrometric analysis to be 15% and have included respective error bars in the Figure 2 in the manuscript.

In terms of the determination of the correction factor we assume that calculating and using an average correction factor based on the individually calculated ones as shown in Table S8 is the best way to account for individual errors in this procedure.

Losses due to degassing: In section S2.1. and section 2.2, it is stated that the reaction solutions were deoxygenated by nitrogen. In section 2.2.1., it is further stated that this led to losses of reagent (NEM).

We carried out a dedicated experiment to assess losses of reagents due to degassing and found that no significant amounts of NEM are lost, see Supplementary Information section 2.2.1.

Why the authors did not use, for instance, freeze-pump-thaw techniques to circumvent this problem?

As we established, there are no significant losses of reagents during deoxygenation with nitrogen gas. Freeze-pump-thaw techniques are more suited to Schlenk-flasks than 0.7 mL crimped glass vials. The small glass vials are not suited to withstand vacuum but are required to perform reactions with controlled photon count both using our tunable laser setup as well as the LED reactor setup.

As the quantum yield is sensitive to the concentration and probably not only NEM but also a part of the solvent was lost, can reproducibility of the results be assumed? How can you exclude that the concentration dependency of the quantum yield is not a methodical error?

We investigated the concentration-dependent quantum yields at 307 nm using NMR spectroscopy, see Figure 2 in the manuscript. The previously observed trends are confirmed.

When we initially carried out the deoxygenation procedure, the loss of solvent did not appear to be quite too significant. Nevertheless, we have now quantified the degree of solvent loss: We have confirmed experimentally (weighing of sample prior and after deoxygenation) that the deoxygenation procedure does not lead to losses of solvent greater than 24 mg acetonitrile (31 μ L, reduction of acetonitrile volume from 0.5 mL to 0.47 mL). We have accounted for this change (other initial solvent volumes as well, see Supplementary Information, section 2.2.2) and adjusted the concentrations.

Are the quantum yield measurements single measurements or were duplications or triplications done?

We now repeated the wavelength-dependent laser experiments for the determination of quantum yields twice more (resulting in triplicate determination of quantum yields), the data is added to the Supplementary Information and the Figure in the manuscript is updated.

Section 2.2.2. In line two, there is a cross reference to section 2.6. regarding mass spectrometry. Section 2.6. contains NMR data. Do you mean section 2.1.?

Yes, this is corrected now.

Minor comments

First paragraph of the introduction: "This paradigm change is not only of academic interest, but has critical connotations for photo-pharmacology and biomedical applications". I do not understand the differentiation between "academic interest" and "photopharmacology and biomedical applications". I consider also the last two content of academic interest as well as synthetic photochemistry content of applied, i.e. industrial interest. Perhaps one should also explicitly mention the major developments currently in photo-redox catalysis.

We thank the reviewer for pointing out that this statement was unclear. We have rephrased it to capture this distinction better. We further make a connection to recent work in the field of photo-catalysis.

Page 9, first paragraph: "[...] not being exactly as good as predicted, the outcome is still noteworthy and could be applied in situations where other chromophores are unsuitable." It is not clear which situations the authors refer to. The selectivity of the system is convincing and the amount of side product probably close to the error margin of the analysis.

One paragraph before the noted sentence, we emphasize that a similar sequence-dependent I-orthogonal ligation system relies on the use of tetrazoles. We highlight that these are challenging to use in 3D direct laser writing due to the formation of a fluorescent adduct and high absorbance of both tetrazole and pyrazoline product. We have added a brief note linking to the previous explanation.

The end of the results: in vivo and ab initio do not need a hyphen

We thank the reviewer for the detailed review, this is corrected now.

Reviewer #3:

Barner-Kowollik and coauthors report a detailed photophysical investigation of all parameters affecting a photocatalytic reaction. They use a self-printed reactor set up to fix the irradiation geometry for LED photoreactions and a tuneable laser set up. A model photoligation was investigated. In addition, computational tools were developed to predict the reaction outcome depending on changes in the reaction parameters, such as the irradiation wavelength. The study is detailed, but in my opinion rather specialized. I have difficulties imagine a broader application of the method in photocatalysis, although better parameter determination and prediction of reaction outcomes would be highly desirable.

We thank the reviewer for the response. Light-induced ligation reactions and photocatalysis are mechanistically quite different. The reaction studied here is not a photocatalytic reaction, the chromophore is the light-absorbing moiety, undergoes intersystem crossing (possibly after internal conversion), a [1,5]-H-shift, bond rotation, likely passing through a conical intersection and finally the Diels Alder Cycloaddition occurs. We expect the method to be directly applicable to other photoligation reactions and photocleavage reactions, but only with possibly some modifications to be applicable to photoinitiators (due to polymerisation mechanisms) and photocatalysis, especially in case of

heterogeneous catalysts. Yet, the concept of a photon-controlled experiment with a 3D-printed reactor could find direct applications in the investigation of photocatalytic systems.

The determination of photophysical parameters in this study uses typical methods. These are available in a spectroscopy lab, but not always in organic synthetic laboratories. The 3-D printed reaction set up is slightly disappointing, as commercial alternatives have been reported with equal or better performance (see for example the photoreactors by Merck and MacMillan, ICIQ start up, www.photoreactor.de). The prediction tool is interesting, but would only be useful for a broader audience if provided as tool with easy interface and input parameters a synthetic lab is able to provide.

The reviewers notes that the methods may be too specialised to find broad application in organic synthesis laboratories. We partly agree with this assessment and have made thorough attempts to make the methods more accessible. Yet, with the specific challenge of precision photochemistry with LEDs comes a set of certain minimal methodical requirements, of which some are unavoidable.

First, the requirement of a tunable laser can be avoided, if only reactions are employed, which have already been investigated accordingly by other research groups. There is a growing number of photoreactions that are being investigated with tunable laser systems, providing a growing literature database of relevant information. With every new publication reporting a precision characterisation of a photochemical reaction with a tunable laser, this information can be used by other laboratories.

Further, the 3-D printed photoreactor in combination with the respective detector scaffold (or a related, modified design, for example one accommodating larger reaction vessels) and a suitable power meter can be used to estimate quantum yields. We have modified the source code (added option to predict conversion based on an arbitrary quantum yield, which allows to determine a quantum yield estimate based on an LED experiment) and included an example calculation (results) in the Supplementary Information, section 2.5. Thereby, the conversion of modified reactions using the same reactor, e.g. addition of a competitive absorber, can be predicted with the algorithm.

Gaining a high performance or efficiency of the photoreactor was not the sole purpose of the design of the 3D-printed reactor. Reproducibility (defined irradiation geometry, scaffolds can be manufactured anywhere in the world to the same specification), versatility (any LED can be integrated into the setup very easily), cost (the scaffold was printed at no significant cost at the additive manufacturing unit at QUT, the detector scaffold was manufactured externally at a cost of 15 Australian Dollars) and precision (light dose can be determined with the detector scaffold) were the central design principles. Importantly, showing that the concept for photoreactor design works, particularly enabling precise control, is a very important message, possibly leading to the future development of related designs with an improved performance and efficiency, using higher power LEDs or larger sample sizes or designs that lead to less loss of light.

Understandably, from the perspective of a start-up that is trying to sell their photoreactor product to laboratories, a publicly available, negligible-cost, precision LED photoreactor could be seen as non-ideal. In the pursuit of conducting science for an international community without barriers to scientific developments, we refrain from patenting or commercially exploiting the innovations reported herein. The opposite is true, we provide both code and stereolithography files, enabling any researcher to reproduce and use our innovative approach free of charge.

We have attempted to make the predictive algorithms more accessible. This includes annotations in the source code itself, to guide the user through applying the code and we have uploaded the code including an explanation on how to apply it to git-hub, an online repository for open-source programs. This is the current standard regarding similar tools. We refer to the programs 'chemcoder' and 'chemreader', published by the group of Filip DuPrez (<https://www.nature.com/articles/s41467-018-06926-3>).

Overall, this rather specialized study showcase that photochemical reactions can be understood and predicted in detail upon careful determination of all parameters. The manuscript is more suitable for a specialized journal focusing on photochemistry. It should be revised to be better accessible for potential users; currently the barrier for an synthetic organic chemist to use the methods is far too high.

We partly disagree in this assessment, especially since the other two reviewers point out that the study is well-suited for the broad audience of Nature Communications. Nevertheless, we have undertaken major changes and made far-reaching additions to allow easier access to the methods that we introduce here.

REVIEWERS' COMMENTS

Reviewer #1 (Remarks to the Author):

The authors have carefully revised this manuscript and answer all the questions. It is a very well written article, which will attract lot of interests. Therefore, I would like to recommend it.

Reviewer #2 (Remarks to the Author):

I went thought the revised version and rebuttal and changes made. I am pleased to see that the authors carefully made the revisions, provided explanations and handled the issues brought forward. I recommend publication of the manuscript in it present form.

Reviewer #3 (Remarks to the Author):

The authors responded in detail to all comments of the reviewers. While I still see a significant barrier to use the described approach in organic synthesis the manuscript may be of value for the photochemical community. Time will tell the use by others and therefore the manuscript should be published.

Point-by-point response to reviewer's comments on manuscript NCOMMS-20-25497A
Predicting Wavelength-Dependent Photochemical Reactivity and Selectivity

Reviewer #1:

The authors have carefully revised this manuscript and answered all the questions. It is a very well written article, which will attract a lot of interest. Therefore, I would like to recommend it.

We thank the reviewer for the positive assessment of our work and for taking the time to carry out a detailed review.

Reviewer #2:

I went through the revised version and rebuttal and changes made. I am pleased to see that the authors carefully made the revisions, provided explanations and handled the issues brought forward. I recommend publication of the manuscript in its present form.

We thank the reviewer for the careful review of our article and the constructive and helpful feedback.

Reviewer #3:

The authors responded in detail to all comments of the reviewers. While I still see a significant barrier to use the described approach in organic synthesis the manuscript may be of value for the photochemical community. Time will tell the use by others and therefore the manuscript should be published.

We thank the reviewer for their critical, yet supportive assessment.

We would like to point out that the use of certain methods, whether 3D printing or the application of algorithms in organic synthesis may at present still seem cumbersome and complicated to many researchers, we are convinced that there already is a change happening with future generations increasingly making use of digital and novel manufacturing technologies with ease. The adoption of new technologies and methods may require a bit of 'activation energy' but could then lead to overall more efficient or precise outcomes, with the potential of becoming a widely adopted standard or leading to more advanced methods, which become state-of-the-art methods.